# Robust Answers, Fragile Logic:
# Probing the Decoupling Hypothesis in LLM Reasoning

**Enyi Jiang, Changming Xu, Nischay Singh, Tian Qiu, Gagandeep Singh**
**University of Illinois at Urbana-Champaign**
`{enyij2,cx23,nischay2,tq8,ggnds}@illinois.edu`

**Reviewed on OpenReview:** `https://openreview.net/forum?id=pMhTFUdM4G`

## Abstract

While Chain-of-Thought (CoT) prompting has become a cornerstone for complex reasoning in Large Language Models (LLMs), the robustness of the generated reasoning remains an open question. We investigate the *Decoupling Hypothesis*: that the robustness of a model's reasoning path and the robustness of its final answer are largely independent; correct answers can coexist with arbitrarily fragile reasoning under small input perturbations. To systematically verify this, we introduce MATCHA, a novel *Answer-Conditioned Probing* framework. Unlike standard evaluations that focus on final output accuracy, MATCHA isolates the reasoning phase by conditioning generation on the model's predicted answer, allowing us to stress-test the stability of the rationale itself. Our experiments reveal a critical vulnerability: under imperceptible input perturbations, LLMs frequently maintain the correct answer while generating inconsistent or nonsensical reasoning - effectively being "Right for the Wrong Reasons". Using LLM judges to quantify this robustness gap, we find that multi-step and commonsense tasks are significantly more susceptible to this decoupling than logical tasks. Furthermore, we demonstrate that adversarial examples generated by MATCHA transfer non-trivially to black-box models. Crucially, we show that this fragility is not solely an artifact of our answer-conditioned protocol: while standard CoT-then-Answer generation does not permit strict answer-fixed isolation, it nevertheless exhibits similar patterns of reasoning degradation under analogous attacks. Our findings expose the illusion of CoT robustness and underscore the need for future architectures that enforce genuine answer-reasoning consistency rather than mere surface-level accuracy.

## 1 Introduction

Large Language Models (LLMs) like GPT-4 (Achiam et al., 2023), Llama-3 (Dubey et al., 2024b), and DeepSeek-R1 (Guo et al., 2025a) have demonstrated remarkable capabilities in complex reasoning through Chain-of-Thought (CoT) prompting (Wei et al., 2022a;b). The prevailing assumption is that these intermediate steps reflect a robust causal process: the model "thinks", and therefore it "answers". However, we challenge the validity of this causal link. We propose the *Decoupling Hypothesis*: the robustness of a model's reasoning path and the robustness of its final answer are largely independent; correct answers can coexist with arbitrarily fragile reasoning under small input perturbations. This hypothesis is most precisely testable under an answer-conditioned protocol where the answer is fixed; in the standard CoT-then-Answer setting, autoregressive coupling prevents strict isolation, though we observe analogous degradation (Section 5.5). While CoT is widely deployed for explanation in education Grassucci et al. (2025) and healthcare Ali (2024), current evaluations fail to distinguish between a model that reasons to reach an answer and one that merely rationalizes a predetermined decision.

To rigorously disentangle these two processes, we introduce a novel *Answer-Conditioned Probing* framework. Unlike standard adversarial attacks that target the final output label, our unique setting isolates the reasoning phase by conditioning the rationale generation on the model's own predicted answer. This allows us to

simulate a critical cognitive failure mode: akin to a student who guesses the correct answer by intuition but hallucinates a derivation to justify it. Importantly, we validate that this fragility is not solely an artifact of the answer-first protocol: although standard CoT-then-Answer generation does not allow strict answer-fixed isolation, it still exhibits similar patterns of reasoning degradation under analogous attacks (Section 5.5). Within this framework, we explicitly attack the rationale generation while the answer remains committed. We demonstrate that imperceptible perturbations can collapse the logical consistency of the reasoning chain without flipping the answer. This phenomenon reveals that models can be "Right for the Wrong Reasons", creating a false sense of security where robust answers mask fragile, inconsistent reasoning.

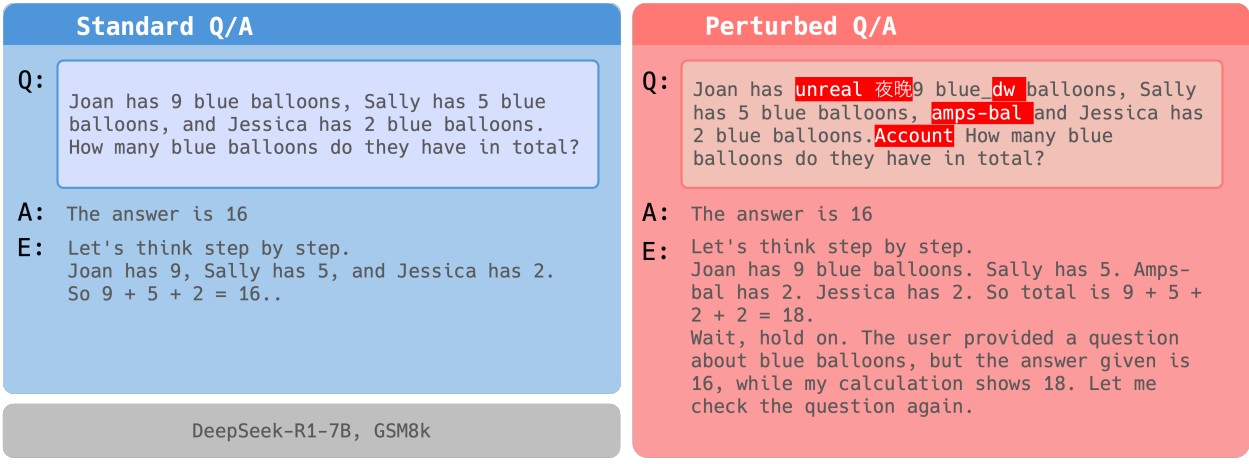

Figure 1: Perturbations in the input question can make reasoning wrong while preserving the correct answer, indicating an underlying problem with answer-reasoning alignment. Example shown using token-level MATCHA applied to DeepSeek-R1-7B on GSM8k.

In this paper, we show the fragility of LLM CoT, as exemplified in Figure 1, where a perturbed question leads to the wrong reasoning yet correct answer (DeepSeek-R1-7B on GSM8K). We adopt an answer-first, then reasoning framework to isolate failures in reasoning while keeping the answer fixed. This setup not only exposes reasoning-specific vulnerabilities but also reflects real-world scenarios where LLM reasoning is often generated post hoc to justify already-decided outputs Jeyasothy et al. (2023); Xu et al. (2024). To this end, we introduce MATCHA, a novel algorithm for finding small input changes that cause inconsistent behavior in CoT reasoning. We use MATCHA to create metrics for evaluating the robustness of an LLM's CoT for different models. MATCHA can be instantiated as either a token-level or embedding-level perturbation. Our primary focus is on **token-level perturbations**, as they are more realistic; however, they are challenging to create due to the need to introduce minimal, semantically plausible edits that selectively disrupt reasoning without altering answer correctness. To achieve this, we first randomly insert tokens into the original question. Then, we identify the inserted tokens that have the largest influence on the reasoning process while ensuring that answer-related tokens remain unchanged. By leveraging gradient-based importance ranking, we selectively replace tokens that maximize reasoning perturbation while minimizing changes to the original question. MATCHA as an **embedding-level perturbation** perturbs the embedding space of input questions imperceptibly, ensuring that the token sequence remains unchanged while disrupting the reasoning process. By optimizing a loss that maximizes reasoning divergence while preserving the correct answer, the perturbation subtly shifts the internal representations, leading to incorrect reasoning without altering the final answer.

**Main contributions**:

- To our knowledge, we are the first to systematically probe this *Decoupling Hypothesis*. By leveraging our *Answer-Conditioned* framework, we quantify the critical misalignment between reasoning stability and answer accuracy. We introduce a novel evaluation framework MATCHA (Misaligned Answer and Thought CHAin) which works at both the token- and embedding-level. We further validate that the observed fragility persists under standard CoT generation.

- We propose an automated robustness assessment framework with a new evaluation metric and LLM-based evaluation system. Our results show that MATCHA significantly lowers the CoT robustness of state-of-the-art models, suggesting that CoTs are fragile and there is misalignment between LLMs' reasoning and answers. Also, we show that LLMs are more vulnerable to perturbations in multi-step and commonsense reasoning scenarios.
- We study the transferability of token-level adversarial examples to black-box LLMs (GPTs), revealing non-trivial transfer rates, underscoring the widespread need for enhancing LLM reasoning robustness and reasoning-answer consistency.

## 2  Related Work

**Prompt-based Reasoning**. Chain-of-Thought was first introduced by Wei et al. (2022a), showing that inducing intermediate reasoning steps significantly improves the reasoning ability of LLMs, which is widely regarded as an important form of NLEs. Many works have further expanded on Chain-of-Thought (CoT) using self-consistency and active prompting methods (Wang et al., 2022; Kojima et al., 2022a; Diao et al., 2023). In addition, Yao et al. (2023) proposes Tree-of-Thought, enabling models to explore multiple reasoning paths for solving complex problems, while Besta et al. (2024) introduces Graph-of-Thought, structuring LLM thoughts as graph vertices to enhance reasoning. We focus on CoT in this paper due to its widespread adoption and relative simplicity, while noting that our framework and methodology remain applicable to other reasoning approaches.

**Sensitivity and stability of explanations.** In the vision domain, Dombrowski et al. (2019); Heo et al. (2019) show that models can easily be fooled into producing wrong explanations with correct predictions. In the language domain, many works have studied the stability (Situ et al., 2021) and faithfulness (Madsen et al., 2024; Chuang et al., 2024; Lanham et al., 2023) of LLM explanations, as well as quantifying uncertainty, counterfactual simulatability, and reliability of explanations (Tanneru et al., 2024; Chen et al., 2023; Ye & Durrett, 2022). We emphasize that our work targets reasoning *robustness* (sensitivity to input perturbations) rather than *faithfulness* (whether explanations reflect the model's true internal process). Compared with their work, MATCHA is the first evaluation framework for measuring the sensitivity of CoTs against token-level and embedding-level perturbations, where we provide a novel angle into the reasoning mechanism and robustness of different models.

**Adversarial attacks**. Adversarial attacks have been extensively studied in traditional machine learning domains such as computer vision (Madry et al., 2018; Goodfellow et al., 2014a; Carlini & Wagner, 2017). In NLP, prior work has explored robustness through synonym substitution, paraphrasing, and character-level perturbations (Jin et al., 2019; Li et al., 2020; Ribeiro et al., 2020; Wang et al., 2021), primarily focusing on inducing prediction flips. More recent studies show that LLMs are highly sensitive to irrelevant context (Shi et al., 2023) and superficial problem rewording (Mirzadeh et al., 2025). In the LLM safety literature, gradient-based jailbreak methods (Zou et al., 2023; Zhu et al., 2023b; Ji et al., 2024; Liu et al., 2024) demonstrate that carefully optimized prefixes or suffixes can bypass alignment safeguards, while Vega et al. (2023) shows that simple accepting phrases can similarly circumvent safety mechanisms. In embedding space, Schwinn et al. (2024) reveals that subtle continuous perturbations can bypass alignment and recover unlearned behaviors, and Xhonneux et al. (2024) further demonstrates the effectiveness of continuous optimization-based attacks against aligned models. Beyond safety-oriented attacks, several works have investigated robustness in reasoning settings. Zhou et al. (2024) studies robustness to noisy rationales, Xu et al. (2024) proposes preemptive-answer CoT attacks without evaluating the faithfulness of the reasoning process itself, and He et al. (2024) leverages CoT prompting to improve robustness on difficult reasoning tasks. Unlike prior approaches that primarily evaluate robustness through final-answer accuracy or attack success rate, MATCHA fixes the final answer and isolates the effect of perturbations on the reasoning trajectory itself. This enables a distinct diagnostic perspective on reasoning robustness, focusing on the stability and consistency of intermediate reasoning rather than only whether the final prediction changes.

**LLM evaluations**. Existing studies evaluate LLMs on accuracy, robustness, and alignment with human values. Benchmarks like GLUE and SuperGLUE (Wang, 2018; Sarlin et al., 2020) assess task performance, while HELM (Liang et al., 2022) provides a holistic evaluation of safety, fairness, and efficiency. Quacer-B (Chaudhary et al., 2024) introduces a certification framework for detecting LLM bias. On the other hand,

LLM-as-a-judge is widely used (Zhu et al., 2023a; Zheng et al., 2023; Li et al., 2024; Huang et al., 2024) in evaluating different capabilities of chat-boxes and LLMs. MATCHA can be used in parallel with these works as it focuses on the novel task of evaluating LLMs on the robustness of their reasoning using LLMs as judges.

## 3  Background

To formally introduce MATCHA, we first cover adversarial attacks and in-context learning.

**Adversarial attacks.** Adversarial examples are a class of robustness attacks on neural networks (Goodfellow et al., 2014b; Kurakin et al., 2018). In image classification, we consider samples $\{(x_i, y_i)\}_{i=0}^N$ from an empirical distribution $\mathcal{D}$, where each image $x \in \mathbb{R}^d$ has a corresponding label $y \in \mathbb{R}^k$. A classifier $f$, parameterized by $\theta$, is trained to minimize a loss function $\mathcal{L} : \mathbb{R}^k \times \mathbb{R}^k \to \mathbb{R}$. Adversarial attacks seek a local point $x'$ within a predefined adversarial region around $x$, i.e. $B_p(x, \epsilon_p) = \{x' \in \mathbb{R}^d : \|x' - x\|_p \le \epsilon_p\}$, such that the classification of $f$ changes ($f(x') \ne f(x) = y$). $x'$ and $x$ should be semantically identical, i.e. a human should determine them as being of the same class. A simple adversarial attack, introduced by Goodfellow et al. (2014b), linearizes the loss function to compute perturbations that maximize the loss while adhering to adversarial constraints, requiring a single backpropagation step:

$$x' = x + \epsilon \cdot \text{sign}(\nabla_x \mathcal{L}(f(x), y))$$

However, gradient-based attacks do not directly apply to text data because 1) the text data space $x$ is discrete instead of continuous, and 2) the adversarial region is hard to define for the text data. In this work, we define two perturbations for optimizing text adversarial examples based on our targeted loss function using gradient information: a) **token-level attack**: random token insertion and gradient-informed replacement using token gradients derived by our novel loss function, b) **embedding-level attack**: imperceptible $l_\infty$ perturbations on question inputs' embedding spaces.

**In-context Learning (ICL) with CoTs.** Our evaluation framework begins by using a set of labeled instances, each paired with a human-crafted CoT, to prompt LLMs. In ICL, given an unlabeled question $x_p \in \mathcal{X}$ and training examples $(x_i, y_i, r_i)_{i=1}^k$, where $x_i \in \mathcal{X}$ is a question, $y_i \in \mathcal{Y}$ is its answer and $r_i \in \mathcal{R}$ is the corresponding CoT, we generate the most likely answer and CoT for the unlabeled question $x_p$ with model $f$ parameterized by $\theta$:

$$\underset{(y_p, r_p) \in \mathcal{R} \times \mathcal{Y}}{\text{argmax}} P_\theta \left( (y_p, r_p) \mid (x_i, y_i, r_i)_{i=1}^k, (x_p) \right)$$

We aim to generate the most probable pair of an answer $y_p$ and a Chain-of-Thought $r_p$ from an LLM. We define $y_p$ as a preemptive answer, meaning that the answer is generated before the reasoning. This design choice allows us to rigorously isolate the robustness of the reasoning process via an *Answer-Conditioned strategy*. In standard reasoning-first approaches, the answer is causally downstream of the rationale; as a result, perturbations that degrade the reasoning often propagate to the final prediction. This coupling introduces a confounding factor that obscures the specific phenomenon of "Right for the Wrong Reasons". By anchoring the model's predicted answer first, we explicitly decouple decision-making from rationalization. This enables us to control the decision boundary and stress-test the reasoning in isolation, ensuring that any observed fragility reflects properties of the reasoning structure itself rather than being a consequence of answer changes.

## 4  MATCHA: Misaligned Answer and Thought CHAin

In this section, we introduce a novel evaluation framework MATCHA that constructs perturbations that selectively degrade CoT quality while preserving the generated correct answer. Unlike traditional perturbations that target model predictions directly, our goal is more nuanced: to expose discrepancies between correct answers and faulty reasoning, thereby revealing hidden failure modes in the reasoning process. Achieving this demands a carefully crafted optimization strategy. To be more specific, we present two complementary methods: **token-level** perturbations, which manipulate the input at the discrete level, and **embedding-level** perturbations, which operate directly in the continuous representation space. Each method presents its challenges and reveals different aspects of the model's vulnerability.

## 4.1 Answer-Conditioned Probing

A key criterion in designing the perturbation is defining its objective, specifically, selecting the loss function that optimizes perturbations to ensure the LLM produces the correct answer while generating incorrect reasoning. The high-level intuition is to *make the CoT different yet preserve the same answer through optimization.* To achieve this, we formulate novel loss functions for our perturbations. We illustrate the basic loss instantiations using an example as shown below:

**ICL** ($n_1$): A list of (Q, CoT, A).

**Q** ($n_2$): If there are 3 cars in the parking lot and 2 more cars arrive, how many cars are in the parking lot?

**A** ($n_3$): The answer is 5.

**CoT** ($n_4$): Let's think step by step. There are 3 cars at first. 2 more cars arrive. $3 + 2 = 5$.

The start location of the in-context example, current question, current answer, and current explanation are denoted as $n_1, n_2, n_3, n_4$, respectively. Further, we define an LLM as a function $f$ that maps a sequence of tokens $x_{1:n}$ to an output logit distribution $f(x_{1:n})$, where tokens $x_i$ belongs to a vocabulary of size $V$. In the following sections, we use logit distributions to help design the loss terms for $\mathcal{L}_c$ and $\mathcal{L}_a$ for the losses on CoT and answer part, respectively. Given a pair of original (reference) response $x$ and perturbed response $x'$, we have $q_c, q_c'$ as the logits of $x_{n_3:n_4}$ (until CoT part), $q_a, q_a'$ as the logits of $x_{1:n_3}$ (until answer part) (Eq. 1). If the logits are computed using the embedding space, then $x$ is an embedding instead of set of tokens. Then, $\mathcal{L}_c$ and $\mathcal{L}_a$ are defined as the cross-entropy over the perturbed distribution compared with the reference distribution.

$$q_c, q_c' \leftarrow f(x_{n_3:n_4}), f(x'_{n_3:n_4}); \ q_a, q_a' \leftarrow f(x_{1:n_3}), f(x'_{1:n_3}) \tag{1}$$

$$\mathcal{L}_c = \mathcal{L}_{CE}(q_c', q_c), \ \mathcal{L}_a = \mathcal{L}_{CE}(q_a', q_a) \tag{2}$$

The optimization objective carefully balances two competing goals: altering the reasoning while preserving the answer. To achieve this, we combine the two loss terms, scaling them via a coefficient $\lambda = \frac{n_4 - n_3}{n_3}$ to give uniform importance to each token. $\mathcal{L}_{opt}$ encourages the model to *maximize* $\mathcal{L}_c$, driving divergence in the CoT, while simultaneously *minimizing* $\mathcal{L}_a$, ensuring the final answer remains intact; thus, precisely targeting the model's reasoning instead of the answer.

$$\mathcal{L}_{opt} = \mathcal{L}_c - \lambda \cdot \mathcal{L}_a \tag{3}$$

## 4.2 Token-Level Perturbations

We design our token-level perturbation strategy to maximize the success rate of MATCHA while maintaining controlled edits to the input. Rather than directly replacing existing tokens, which risks distorting the original question and degrading answer accuracy, we adopt a two-stage token insertion and replacement approach. This allows us to inject perturbations without overwriting semantically critical content, preserving the model's ability to produce the correct answer. We then refine these inserted tokens through gradient-guided replacement, identifying which modifications most effectively disrupt the reasoning while keeping the answer stable. Compared to existing methods like GCG, our approach is more targeted and effective, yielding a lower unattackable rate and higher success rates (see Table 3). This design reflects a core insight: disrupting reasoning does not require semantic collapse, just precise, minimal shifts that steer the model's internal trajectory off-course.

**Stage 1: Random token insertion.** We initiate our attack by introducing a set of randomly selected tokens into the original question at randomly selected positions, effectively simulating a perturbation to the input, which we denote as "_" in Example 4.2. These random tokens serve as an initial disturbance that subtly alters the original question while preserving its structure and intent. The insertions aim to create a minimally modified version of the question that can expose vulnerabilities in the model's reasoning or

decision-making process. We denote the ratio of inserted tokens compared to the original question length $(n_2 - n_1)$ as $a$.

**Stage 2: Gradient-informed token replacement.** We formulate our perturbation as an optimization problem by replacing the inserted tokens that are important for *maximizing $\mathcal{L}_c$ while minimizing $\mathcal{L}_a$*, to induce incorrect reasoning with the correct answer. First, we generate $y_p$ and $r_p$ by passing the unlabeled question $x_p$ into the LLM. If $y_p$ equals $y_{gt}$ (ground truth label), we then generate reference logits $q_c, q_a$ (Eq.1). After that, we start the optimization by generating the perturbed logits $q'_c, q'_a$ (Eq.1) by passing in the perturbed inputs $x'$. With Eq. 2, we calculate $\mathcal{L}_c$, $\mathcal{L}_a$ as well as $\mathcal{L}_{opt}$.

We compute the gradient at each one-hot token indicator $e_x$ with a shape of $((n_2 - n_1) * a, |V|)$, which has a value of 1 at the current input id and 0 elsewhere. To identify promising replacement candidates for each inserted token position, we evaluate them via forward passes. Specifically, we approximate the effect of replacing token $x_i$ using its gradient, as shown in Eq. 4, where $g_{tok}$ has a shape of $((n_2 - n_1) * a, |V|)$. We use this to measure the influence of each token on the reasoning/answer. However, just knowing the influence is not enough; we need to use $g_{tok}$ to find the *right locations* for the replacements and *right replacements* for those selected locations.

$$g_{tok} = \nabla_{e_x}(-\mathcal{L}_{opt}) \tag{4}$$

To find a set of *right locations* for inserted tokens (in the question part $x_{n_1:n_2}$), we compute $g_{tok} \cdot e_x$ (the gradient of the current token at each position). We then select the top-$k$ token positions (highlighted in $\boxed{\text{yellow}}$) with the highest gradient scores (i.e. the positions that are most important for encouraging answer-reasoning inconsistency), as shown in Eq. 5, where $Q$ denotes the selected token indices for replacement.

$$Q = \text{TopK}(g_{tok} \cdot e_x, k) \tag{5}$$

To find the *replacement token* for each selected position $q \in Q$, we identify the replacement token by selecting the vocabulary index with the highest gradient magnitude (Eq. 6).

$$rank_{tok}[q] = \arg\max_{v \in [|V|]} g_{tok}[q, v] \tag{6}$$

Finally, we get the replacement token list $rank_{tok}$, where each element corresponds to the top-ranked replacement token for each promising position. Specifically, the token that is predicted (via gradient guidance) to most effectively perturb the model's reasoning while preserving the correct answer.

> **Q** ($n_2$): If _ there are _3 cars_ in the _ parking lot and _ 2 more _ cars _ arrive, _ how many _ cars are _ in the _ parking lot?

For each token in $rank_{tok}$, we leverage an LLM judge with inputs of original/perturbed questions to detect whether the question has a similar semantic meaning after replacing the token (the judge template can be found in Table 20). If the meaning changes, then we do not replace that token. We continue iterating this replacement/judge process for $j$ steps. Our final perturbed answer $y'_p$ and CoT $r'_p$ are generated with the final perturbed input $x'_p$. We check the reasoning correctness of $r'_p$ by passing $(r_p, r'_p)$ into another judging LLM (the judge template can be found in Table 19). Our perturbation is successful if $y'_p$ is correct yet $r'_p$ is judged wrong. We present our token-level perturbation procedure in Algorithm 2 and 3.

### 4.3 Embedding-Level Perturbations

Embedding-level perturbations provide a powerful lens into LLM vulnerabilities beyond the discrete token space Schwinn et al. (2024); Xhonneux et al. (2024). Thus, we leverage embedding-level perturbations to subtly distort the model's reasoning path while preserving answer correctness, exposing latent instabilities that are not easily revealed by token-level edits. The high-level idea is to perturb the embedding space of the input questions ($E_c[n_1 : n_2]$) imperceptibly, such that it does not change the output token mapping, leading to the wrong reasoning yet correct answer. Similar to token-level perturbations, we generate the predicted answer $y_p$, $r_p$, and check whether $y_p$ equals $y_{gt}$. If so, we then generate target logits $q_c, q_a$ using input

embedding $E_c$. After that, we generate the perturbed logits $q'_c, q'_a$ by passing in the perturbed embedding inputs for CoT and answer parts ($E'_c$ and $E'_c[: n_3]$). Given step-size $\alpha$, perturbation region $\epsilon$ (a percentage of the original embedding space), and using Eq. 3 to calculate $\mathcal{L}_{opt}$, we update the embedding by performing $l_\infty$ perturbation as follows:

$$E_{tmp} \leftarrow E'_c + \alpha \cdot \epsilon \cdot \text{sign}(\nabla_{E'_c} \mathcal{L}_{opt}) \tag{7}$$

Then, we clamp $E_{tmp}$ into the interval of $[E_c - \epsilon, E_c + \epsilon]$ and only update the $E'_c[n_1 : n_2]$ part of the embedding with $E_{tmp}$, since we only perturb the embedding space of the question part. We continue the iterations for the perturbation with a budget of $j$ steps. After that, we generate the perturbed answer $y'_p$ and CoT $r'_p$ with perturbed input embedding $E'_c$. To know whether the perturbation is successful, we follow the same procedure as the token-level perturbations. Algorithm 1 illustrates the embedding-level perturbation procedure of one unlabeled question.

## 5 Experimental Evaluation

| | ACC (%) | | | | UR (%) | | | | SR (%) | | | |
|---|---|---|---|---|---|---|---|---|---|---|---|---|
| | SE | GK | SA | AVG | SE | GK | SA | AVG | SE | GK | SA | AVG |
| Llama-3-8B | 80.5 | 16.1 | 68.0 | 54.9 | **68.9** | 39.0 | 60.3 | **56.1** | **10.2** | 14.7 | 16.2 | **13.7** |
| Mistral-7B | 77.6 | 11.4 | 64.0 | 51.0 | 56.3 | 22.5 | 59.1 | 46.0 | 3.6 | **17.9** | 14.1 | 11.9 |
| Zephyr-7B-beta | 70.9 | 9.9 | 65.8 | 48.9 | 38.9 | 13.8 | 63.2 | 38.6 | 8.1 | 12.3 | 13.4 | 11.3 |
| Qwen2.5-7B | **85.0** | **24.8** | **74.2** | **61.3** | 56.0 | **38.5** | **67.9** | 54.1 | 2.1 | 4.0 | 4.9 | 3.7 |
| DeepSeek-R1-7B | 84.1 | 22.5 | 56.2 | 54.3 | 54.8 | 28.3 | 43.4 | 42.2 | 5.2 | 9.1 | **18.9** | 11.1 |

Table 1: Token-level perturbation results on SingleEQ (SE), GSM8K (GK), and StrategyQA (SA) datasets. We report the accuracy (ACC), unattackable rate (UR), and success rate (SR) for five models across three datasets. The results reveal that the models have a significant sensitivity to the token-level perturbations.

| | ACC (%) | | | | UR (%) | | | | SR (%) | | | |
|---|---|---|---|---|---|---|---|---|---|---|---|---|
| | SE | GK | SA | AVG | SE | GK | SA | AVG | SE | GK | SA | AVG |
| Llama-3-8B | 80.5 | 16.1 | 68.0 | 54.9 | 24.9 | 41.8 | 21.2 | 29.3 | **5.1** | 10.8 | 29.7 | **15.2** |
| Mistral-7B | 77.6 | 11.4 | 64.0 | 51.0 | **88.3** | **63.6** | **79.7** | **77.2** | 0.0 | **18.5** | 2.8 | 7.1 |
| Zephyr-7B-beta | 70.9 | 9.9 | 65.8 | 48.9 | 70.6 | 54.6 | 77.2 | 67.5 | 2.2 | 13.1 | 2.1 | 5.8 |
| Qwen2.5-7B | **85.0** | **24.8** | **74.2** | **61.3** | 1.9 | 46.8 | 18.3 | 22.3 | 0.5 | 1.8 | 24.0 | 8.8 |
| DeepSeek-R1-7B | 84.1 | 22.5 | 56.2 | 54.3 | 20.6 | 59.3 | 10.0 | 30.0 | 6.1 | 6.4 | **33.1** | **15.2** |

Table 2: Embedding-level perturbation results on SingleEQ (SE), GSM8K (GK), and StrategyQA (SA) datasets. We report the accuracy (ACC), unattackable rate (UR), and success rate (SR) for five models across three datasets. The results reveal that the models have a significant sensitivity to the input perturbations, like embedding-level perturbations.

### 5.1 Experimental setup

**Datasets.** We use three datasets in line with existing literature on reasoning (Wei et al., 2022b; Trivedi et al., 2022; Kojima et al., 2022b; Diao et al., 2023). For math reasoning, we select two math datasets SingleEq (Koncel-Kedziorski et al., 2016) with single-step math problems and GSM8K (Cobbe et al., 2021) with multi-step math problems. Apart from that, we select StrategyQA (Geva et al., 2021) for commonsense reasoning. For all datasets, we use the test split for the evaluation. Specifically, we sampled 500 questions for the evaluation of the StrategyQA dataset and used all data (508 for SingleEq and 1319 for GSM8K) in the other two datasets. More details about the datasets can be found in Appendix A.2.

**Models.** Our experiments focus on open-source and closed-source models for robustness evaluation. For open-source models, we use Llama-3-8B (Dubey et al., 2024a), Mistral-7B (Jiang et al., 2023), Zephyr-7B-beta (Tunstall et al., 2023), Qwen2.5-7B (Yang et al., 2024), and DeepSeek-R1-Distill-Qwen-7B (Guo et al., 2025b). We evaluate the transferability of white-box perturbations to closed-source models GPT-4 and GPT-3.5-turbo (Achiam et al., 2023). Also, we use GPT-3.5-turbo as our judging model for comparing CoTs, and DeepSeek-R1-Distill-Qwen-7B (Guo et al., 2025b) for evaluating the semantic meanings of perturbed questions. Human evaluations of those judges are available in Appendix A.4 and Appendix A.7.

**Metrics.** For all datasets, we report the accuracy (ACC), attack success rate (SR), and unattackable rate (UR). ACC represents the percentage of correctly answered questions without perturbations; SR represents the

percentage of problems answered with wrong reasoning when the answers are correct after the perturbation; UR indicates the percentage of problems answered correctly with correct reasoning steps when the answers are correct after the perturbation. WR=1-UR-SR represents the percentage of questions answered correctly before but answered with wrong answers after the perturbation, which is common in our settings. Also, UR is used to represent the robustness of reasoning perturbations (e.g., a higher UR means a higher robustness).

**Implementation.** For token-level perturbation on open-source models, we set the inserted ratio to be $a = 0.2$ for all datasets; for the replacement ratio $k$, we set it to be $(0.5, 0.25, 0.5)$ for SingleEq, GSM8K, and StrategyQA datasets, respectively. For embedding-level perturbation, we choose hyperparameters with perturbation percentage $\epsilon = (0.02, 0.005, 0.03)$ for SingleEq, GSM8K, and StrategyQA datasets, which are imperceptible for the token mapping. Also, we set the number of perturbation steps $j$ to be 5 for all datasets and two kinds of perturbations. For closed-source model experiments, we set temperature $T = 0.7$ and we test the closed-source models on the adversarial examples that open-source models generate. The default version of GPT-3.5-turbo used is GPT-3.5-turbo-0125, and the GPT-4o used is GPT-4o-2024-08-06. Apart from that, we set the maximum length of generated output to be 256. We use a conventional ICL setting for our experiments (Brown et al., 2020; Rae et al., 2021). We apply the same number of exemplars as Wei et al. (2022b) and use the exemplars from Diao et al. (2023), which are 8 for SingleEq and GSM8K, as well as 6 for StrategyQA. To validate our LLM-based judge, we conducted a human evaluation on 50 randomly sampled examples per dataset, finding agreement rates of 74% (SingleEq), 88% (GSM8K), and 78% (StrategyQA) between human annotators and the GPT-3.5 judge (Appendix A.7), consistent with prior work (Zheng et al., 2023). Runtime, LLM judge analysis, and limitations can be found in Appendix A.5 and A.7.

**Evaluation Subset.** By design, MATCHA focuses on probing the "right for the wrong reasons" phenomenon on instances where the model produces the correct answer; for incorrectly answered questions, the reasoning is already unreliable and thus less informative for this analysis. As a result, our evaluation is conducted on the subset of correctly answered examples, whose size varies across models and datasets (e.g., from ∼10% on GSM8K for weaker models to ∼85% on SingleEq for stronger models). This variation primarily reflects differences in underlying model performance rather than our evaluation protocol. To assess whether this subset selection introduces bias, we additionally conduct answer-anchored experiments (Table 11) that evaluate all data points by fixing the answer. The results show consistent and even stronger reasoning fragility under this setting, suggesting that our original subset-based evaluation is conservative and does not drive the observed phenomenon.

## 5.2 Comparison with baselines

In Table 3, we compare MATCHA with the following baselines for token-level perturbations: (1) **Random**: we randomly insert tokens into the original question; (2) **GCG**: we optimize a suffix similar to Zou et al. (2023) using our defined loss; (3,4,5) **MATCHA**: we experiment with MATCHA with random locations/with only CoT loss/full versions. We show that MATCHA can achieve a comparatively lower unattackable rate and higher success rate, indicating MATCHA token-level perturbations have stronger perturbation effectiveness.

## 5.3 Evaluation of open-source models

In Table 1 and Table 2, we show that MATCHA successfully examines and reveals the sensitivity of different models against embedding-level and token-level perturbations across three datasets (two math datasets and one commonsense reasoning dataset). We analyze the performance of our evaluation framework as follows.

|  | UR | SR |
|---|---|---|
| Random | 49.1(0.7) | 10.0(0.6) |
| GCG | 65.4(7.6) | 12.7(6.2) |
| MATCHA (random locs) | **36.0**(7.5) | 12.2(5.0) |
| MATCHA (only $\mathcal{L}_c$) | 41.5(2.9) | 13.0(2.4) |
| MATCHA | 39.0(5.0) | **14.7**(2.4) |

Table 3: Baseline comparisons on GSM8K dataset using Llama-3-8B.

**Embedding-level v.s. token-level attacks.** The relative strength of perturbations at the embedding-level versus the token-level varies between models. For Mistral-7B and Zephyr-7B-beta, token-level perturbations yield lower UR, indicating stronger degradation of reasoning robustness. In contrast, for Llama-3-8B, Qwen2.5-7B, and DeepSeek-R1-7B, embedding-level perturbations are more effective, leading to greater reductions in UR. SR tends to be comparable or slightly higher for embedding-level perturbations, yet the non-trivial SR indicates the effectiveness of MATCHA.

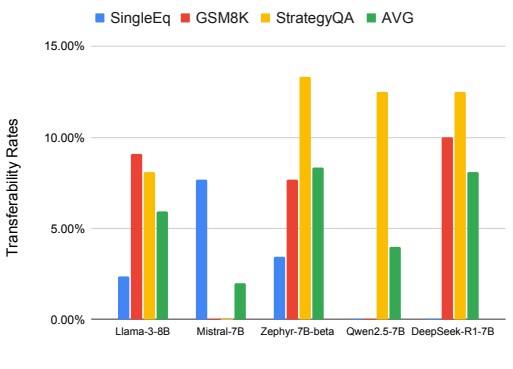 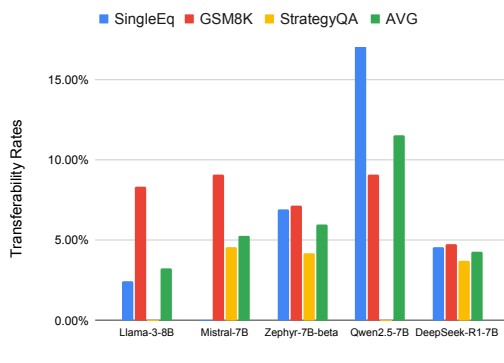

(a) Transferability to GPT-3.5-turbo.        (b) Transferability to GPT-4o.

Figure 2: Transferability to closed-source models (GPT-3.5-turbo and GPT-4o) experiments using the token-level successful examples, showing non-trivial transfer rates from the open-source models.

We observe that both kinds of perturbations reveal different aspects of model vulnerabilities with different behaviors across models.

**Comparison of models.** Mistral-7B and Zephyr-7B-beta show stronger robustness to embedding-level perturbations, with high UR, but are more vulnerable to token-level perturbations, suggesting their reasoning may be more sensitive to discrete token changes. In contrast, Llama-3-8B, Qwen2.5-7B, and DeepSeek-R1-7B are more affected by embedding-level perturbations, exhibiting greater drops in UR, which indicates a greater sensitivity to fine-grained perturbations in the input space. Further, Llama-3-8B and DeepSeek-R1-7B have the relatively highest SR, displaying the most severe inconsistency between reasoning and answer; Qwen2.5-7B has few inconsistent behaviors.

**Comparison of datasets.** Models exhibit lower robustness on multi-step reasoning (GSM8K) compared to single-step questions (SingleEq), likely because MATCHA can disrupt intermediate steps, leading to complete failure. Since multi-step reasoning depends on sequential logic, small perturbations can have a cascading effect, making these tasks more vulnerable. Models show moderate/low robustness on commonsense reasoning (StrategyQA), suggesting implicit knowledge is more vulnerable to perturbations than structured reasoning. While models handle logical reasoning, they remain susceptible to subtle perturbations in knowledge-based inference. Overall, multi-step and commonsense reasoning is more fragile under MATCHA, and improving both structured and knowledge-based answer/reasoning consistency is essential. We note that scaling from 7B to 14B substantially reduces SR on math tasks but leaves commonsense reasoning vulnerable (Appendix A.4).

**ACC v.s. UR.** The accuracy-UR trade-off is more pronounced under embedding-level perturbations: high ACC models like Qwen2.5-7B and DeepSeek-R1-7B exhibit significantly lower UR, while lower-accuracy models such as Mistral-7B and Zephyr-7B-beta are more resilient. This suggests that accuracy-optimized models may develop sharper decision boundaries or more fragile internal representations, making them more vulnerable to fine-grained embedding perturbations. In contrast, token-level perturbations show a less consistent pattern, possibly because discrete substitutions interact with model tokenization in architecture-specific ways, leading to irregular or less predictable robustness degradation. These results highlight the importance of evaluating both perturbation types when analyzing the robustness of LLM reasoning.

### 5.4 Transferability to closed-source models

Figure 2 illustrates the transferability rates of our token-level perturbations to closed-source models GPT-3.5-turbo and GPT-4o. The results indicate non-trivial transferability, suggesting that successful perturbations generated for open-source models can still effectively fool black-box models.

**Model-level analysis**: Qwen2.5-7B produces notably high transferability to GPT-4o, especially on structured tasks like SingleEq and GSM8K, indicating that its perturbations align closely with GPT-4o's decision boundaries. Zephyr-7B and Mistral-7B show strong transferability to GPT-3.5-turbo on different tasks, suggesting that base model capabilities and reasoning biases shape adversarial generalization.

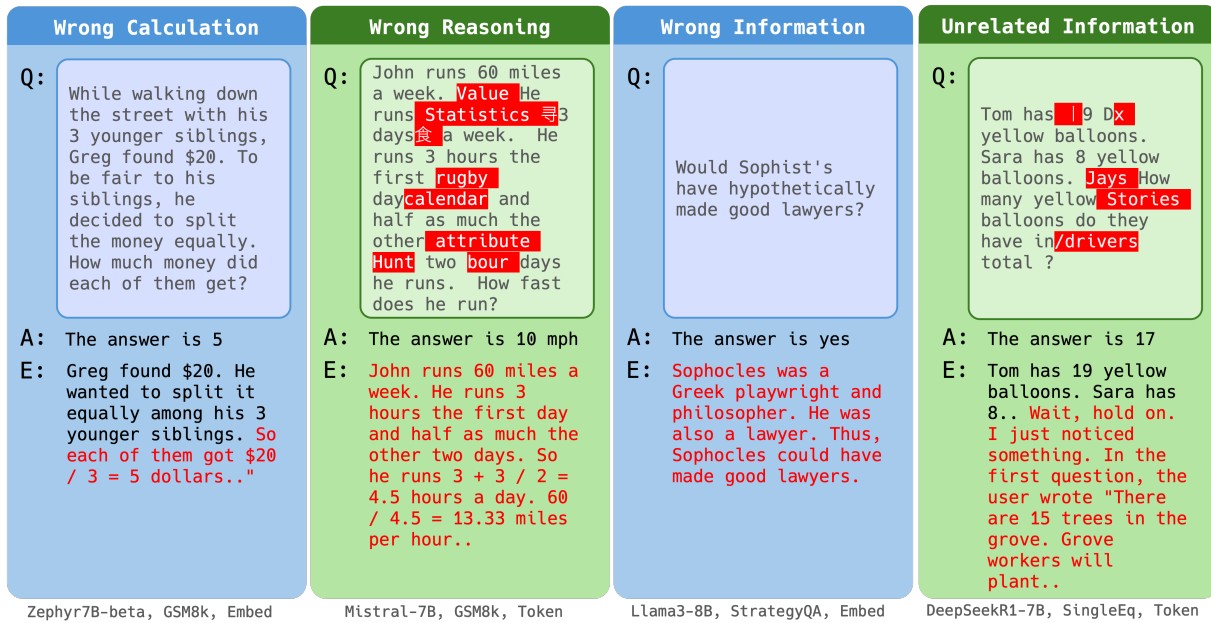

Figure 3: Success examples of our token-level and embedding-level perturbations on different models. We classify the errors into four categories. For token-level perturbations, the replaced tokens are colored in red, and for the CoTs, the wrong steps are colored in red.

| | ACC (%) | | UR (%) | | SR (%) | |
|---|---|---|---|---|---|---|
| **Model** | **C→A** | **A→C** | **C→A** | **A→C** | **C→A** | **A→C** |
| Llama-3-8B | **75.6** | 16.1 | **55.3** | 41.8 | **17.5** | 10.8 |
| Mistral-7B | **45.4** | 11.4 | **63.9** | 63.6 | **17.2** | 18.5 |
| Zephyr-7B-beta | **43.8** | 9.9 | 42.9 | **54.6** | **24.7** | 13.1 |
| Qwen2.5-7B | **60.4** | 24.8 | **60.3** | 46.8 | **6.3** | 1.8 |
| DeepSeek-R1-7B | **85.2** | 22.5 | **80.8** | 59.3 | **7.7** | 6.4 |

Table 4: Comparison between CoT→Answer and Answer→CoT on GSM8K (values in %).

**Dataset-level analysis**: Tasks like StrategyQA exhibit high transferability to GPT-3.5-turbo but much lower transferability to GPT-4o, suggesting that commonsense reasoning attacks transfer less effectively to more advanced models like GPT-4o. In contrast, GSM8K shows more consistent transferability, likely due to its focus on multi-step math reasoning, which may yield more structurally generalizable perturbations.

These results underscore that both types of reasoning in the dataset and the characteristics of the base model jointly determine transferability.

### 5.5 Comparison of CoT-first vs answer-first settings

Evaluating standard CoT (CoT-then-Answer) prompting is necessary to determine whether the observed fragility is specific to the answer-conditioned setup. To achieve this, we implemented a CoT-then-Answer baseline with the same embedding-level perturbations. In this setting, it is impossible to explicitly fix or control the position of the generated final answer due to the autoregressive nature of LLMs. To approximate answer consistency, we identify the answer from the final portion of the generated sequence (e.g., the last quarter of tokens) and measure answer similarity based on this segment. Despite this constraint, we observe that reasoning instability persists in the standard CoT setting. Below (Figure 4) are representative examples where small perturbations lead to clearly degraded or incoherent reasoning, even though the model attempts to follow a step-by-step format.

---

**Model: Llama-3-8B**

**Question (Q):** Carl has a cane that is half as long as he is tall. Carl is one foot taller than his brother, Ned. And Ned is two feet shorter than his cousin, Isabel. If Isabel is 7 feet tall, how long is Carl's cane, in feet?

**Perturbed Rationale (A'):** Let's think step by step. **Isabel is://://.swing Isabel is the the.. the.** Carl is the **taller than Ned by 7 feet**. Ned is **1 foot taller than Carl**.. So Carl is 6 feet tall.. Carl's cane is half as long as he is tall.. So it is 3 feet long. **The answer is 3.**

---

**Model: Zephyr-7B-beta**

**Question (Q):** The vending machines sell chips for 40 cents and candy bars for 75 cents. George spent $5 and got 3 bags of chips and had 1% of his money left. How many candy bars did he buy?

**Perturbed Rationale (A'):** Let's think step by step. George spent 5 dollars. 3 bags of chips cost 3 x 40 = 120 cents. So he spent 120 cents on chips. That leaves **5 - 120 = 380 cents** for candy bars and the remaining money. 1% of 5 dollars is 5 cents. So George has 380 - 5 = 375 cents left for candy bars. 375 cents is 5 candy bars at 75 cents each. So George bought 5 candy bars. **The answer is 5.**

---

Figure 4: Qualitative examples of successful attacks where models exhibit severe logic breakdowns (highlighted in **red**) but still forcefully generate the correct final answer, demonstrating the fragility of CoT.

Importantly, due to the autoregressive nature of LLMs, reasoning and answer generation are tightly coupled in the CoT-then-Answer setting. As a result, when the reasoning chain diverges, the final answer often becomes incorrect simultaneously. This makes it difficult to isolate reasoning-specific failures using answer correctness alone, and consequently the decoupling evidence from this setting is corroborative rather than definitive. To account for this, we adapt our attack success criterion in the CoT-then-Answer setting to focus on reasoning quality, evaluated using the same GPT-based judge.

In this setting, we determine answer correctness by extracting the final numerical or yes/no answer from the last portion of the generated sequence and checking exact match against the ground truth. While this provides a practical way to assess reasoning instability under standard CoT prompting, we acknowledge that developing more precise methods to disentangle reasoning and answer generation in this setting remains an important direction for future work. Under this criterion, we observe non-trivial attack success rates across all models on GSM8K, as shown in Table 4.

Notably, the CoT→Answer setting yields substantially higher ACC (e.g., 75.6% vs 16.1% for Llama-3-8B), meaning the evaluation pool is much larger, which makes the observed SR values even more significant as they reflect fragility across a broader set of correctly answered instances.

### 5.6 Examples of successful attacks

We categorize the successful attack examples into the following four types of errors and select four examples (one per category) across different models/datasets in Figure 3.

**Wrong calculations** refers to the errors in the calculation steps. As shown in the first row of Figure 3, the model thinks "20/3=5", which is incorrect in the calculation. **Wrong reasoning** refers to the evident reasoning errors in the procedure of arriving at the correct solution. For the second example in Figure 3, the model makes the mistake of getting the total number of running hours (should be $3 + 2/3 * 2 = 6$ instead of 4.5), leading to the overall error in the reasoning procedures. **Wrong information** refers to the model outputting some information that is wrong according to the question, to arrive at the correct answer. We present an example in the third column of Figure 3, where the model makes up "Sophocles" instead of "Sophist" to arrive at the correct answer. **Unrelated information** refers to the model outputting some unrelated information when trying to answer the question. In the fourth column of Figure 3, the model starts

to make up non-existent information like "There are 15 trees in the grove" to guess the answer. The complete list of successful attack examples can be found in Appendix A.6.

**On perturbation naturalness.** We acknowledge that some token-level perturbations can appear unnatural to human readers (e.g., inserted subword tokens or non-English characters). However, we emphasize two points: (1) our LLM-based semantic judge filters out perturbations that alter the question's meaning, and human evaluation confirms high semantic preservation rates (98% on SingleEq, 94% on GSM8K, 88% on StrategyQA; Table 5). Also in Table 13, we verify this using BERT-Score (Zhang et al., 2019), which shows high semantic similarity (around 93% across datasets), indicating that the perturbed inputs are close paraphrases rather than meaning-altering edits; (2) the goal of our perturbations is not human-imperceptibility but rather to probe model-level sensitivity: even these small, filtered perturbations suffice to break reasoning while preserving the answer. For embedding-level perturbations, the perturbations are by construction imperceptible at the token level.

Additional ablation studies and analysis can be found in Appendix A.4.

## 6 Conclusion

In this work, we empirically examine the *Decoupling Hypothesis*, questioning the assumption that correct answers in LLMs are accompanied by robust reasoning. Using our proposed MATCHA framework, we introduce *Answer-Conditioned Probing* to disentangle the robustness of a model's prediction from that of its justification. We find that LLMs can often produce correct answers despite fragile or inconsistent reasoning. In particular, small, imperceptible perturbations can disrupt the logical structure of a Chain-of-Thought while leaving the final answer unchanged, especially in multi-step and commonsense tasks. The transferability of these effects to black-box models (e.g., GPT-4) suggests that this phenomenon is not model-specific, but reflects a broader limitation of current training and alignment practices. We note that reasoning fragility under perturbation does not necessarily imply unfaithfulness: a model may genuinely reason via CoT yet have that reasoning be sensitive to noise. Our results establish, most rigorously under answer-conditioned probing and corroboratively under standard CoT generation, that reasoning robustness is decoupled from answer robustness, which is a necessary (though not sufficient) condition for questioning faithfulness. Overall, MATCHA provides a diagnostic perspective on reasoning robustness, highlighting the gap between answer correctness and process reliability. Our results suggest that improving trustworthiness will require moving beyond answer-level supervision toward methods that better couple reasoning structure with model predictions.

## 7 Broader Impacts

Our work highlights that reasoning explanations in LLMs can be more fragile than final answers under small perturbations. While intended as an evaluation and auditing framework, our approach may have dual-use implications. In particular, it could be misused to produce misleading yet plausible post-hoc rationales, especially in high-stakes domains where users may over-rely on model-generated explanations. Additionally, imperceptible perturbations could be exploited to manipulate reasoning outputs subtly. At the same time, our findings emphasize the need for caution when interpreting free-form explanations as evidence of faithful reasoning. To mitigate risks, we encourage reducing reliance on unconstrained CoT, incorporating more verifiable intermediate representations, and adopting robustness-aware evaluation protocols.

## 8 Acknowledgement

We sincerely thank the anonymous reviewers and the action editor for their insightful comments and constructive suggestions. This work was supported by NSF Grants No. CCF-2238079, CCF-2316233, CNS-2148583, Google Research Scholar Award, and an Open Philanthropy research grant.

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

# A Appendix

---

**Algorithm 1** Embedding-level Perturbations

---

**Require:** Number of attack steps $k$, ICL examples $I$ with length $n_1$, unlabeled question $x_p$ with length $n_2$, ground truth $y_{gt}$, model $f$ parameterized by $\theta$, attack region $\epsilon$, stepsize $\alpha$, loss weight $\lambda$, embedding layer $h$, judge LLM $J$.
**Ensure:** Correct $a$, Success $b$, Unattackable $c$, Wrong $d$.

1: Initialize $a, b, c, d \leftarrow 0, 0, 0, 0$
2:                  ▷ Initialization for conditions
3: $(y_p, r_p) \leftarrow \arg\max P_\theta((y_p, r_p)|I, (x_p))$
4:                  ▷ Generate original answer and CoT
5: **if** $y_p == y_{gt}$ **then**
6:      $a \leftarrow 1$
7:      $E_c \leftarrow h(I, (x_p, y_p, r_p))$
8:      $E_a \leftarrow h(I, (x_p, y_p))$                  ▷ Get embeddings
9:      $q_a, q_c \leftarrow f(E_a), f(E_c)$                  ▷ Get logits
10:     $E'_c \leftarrow E_c$
11:     **for** $i \leftarrow 1$ to $k$ **do**
12:         $q'_c, q'_a \leftarrow f(E'_c), f(E'_c[: n_3])$
13:         $\mathcal{L}_a, \mathcal{L}_c \leftarrow L_{CE}(q'_a, q_a), L_{CE}(q'_c, q_c)$
14:         $\mathcal{L}_{opt} \leftarrow \mathcal{L}_c - \lambda \cdot \mathcal{L}_a$             ▷ Loss objective
15:         $E_{tmp} \leftarrow E'_c + \alpha \cdot \epsilon \cdot \text{sign}(\nabla_{E'_c} \mathcal{L}_{opt})$
16:         $E_{tmp} \leftarrow \text{clamp}(E_{tmp}, E_c - \epsilon, E_c + \epsilon)$
17:         $E'_c \leftarrow \text{concat}(E_c[: n_1], E_{tmp}[n_1 : n_2], E_c[n_2 :])$     ▷ Update perturbed embedding
18:     **end for**
19:     $y'_p, r'_p \leftarrow \arg\max P_\theta((y'_p, r'_p)|E'_c[: n_2])$
20:     **if** $y'_p \neq y_{gt}$ **then**
21:         $d \leftarrow 1$, **Break**
22:     **else**
23:         **if** $J(r'_p, r_p) == 0$ **then**
24:             $b \leftarrow 1$, **Break**
25:         **end if**
26:     **end if**
27:     **if** $i == k$ **then**
28:         $c \leftarrow 1$
29:     **end if**
30: **end if**
31: **return** $a, b, c, d$                             ▷ Final conditions

---

## A.1 Limitations

Our method requires significant computational resources due to the complexity of generating and evaluating successful examples on LLMs. The effectiveness of our approach relies on the LLM judge's ability to assess the correctness of reasoning within CoT explanations. However, this evaluation process is not entirely reliable, as LLMs may exhibit inconsistencies or biases in their judgments, which could potentially affect the accuracy of our robustness analysis. Further, the token-level attack may slightly impact the original meaning of the questions.

## A.2 Details of datasets

**SingleEq (Koncel-Kedziorski et al., 2016)** is a collection of algebraic word problems designed to evaluate the mathematical reasoning capabilities of machine learning models. Each problem in the dataset consists of a short natural language description corresponding to a **single-variable linear equation**, making it a benchmark for assessing symbolic reasoning and arithmetic problem-solving skills in language models.

---

**Algorithm 2** Token-level Perturbations

---

**Require:** Number of attack steps $k$, ICL examples $I$ with length $n_1$, input ids and input tokens, unlabeled question $x_p$ with length $n_2$, ground truth $y_{gt}$, model $f$ parameterized by $\theta$, embedding layer $h$, selected number of token $T_k$, judge LLM $J_1$ for reasoning correctness and $J_2$ for question correctness.

**Ensure:** Correct $a$, Success $b$, Unattackable $c$, Wrong $d$.

1: Initialize $a, b, c, d \leftarrow 0, 0, 0, 0$
2:                                           ▷ Initialization for conditions
3: $(y_p, r_p) \leftarrow \arg\max P_\theta((y_p, r_p)|I, (x_p))$
4:                                     ▷ Generate original answer and CoT
5: **if** $y_p == y_{gt}$ **then**
6:     $a \leftarrow 1$
7:     $q_a, q_c \leftarrow f(I, (x_p, y_p)), f(I, (x_p, y_p, r_p))$                  ▷ Get target logits
8:     $x'_p, r'_p \leftarrow x_p, r_p$
9:     $x'_r \leftarrow \text{random\_insert}(x'_p)$           ▷ Randomly insert tokens into the question part
10:    $y'_p, r'_p \leftarrow \arg\max P_\theta((y'_p, r'_p)|x'_r)$
11:    **if** $y'_p \neq y_{gt}$ **then**
12:       $d \leftarrow 1$, **Break**
13:    **else**
14:       **if** $J_1(r'_p, r_p) == 0$ **then**
15:          $b \leftarrow 1$, **Break**
16:       **end if**
17:    **end if**
18:    $\text{insert\_indices} \leftarrow \text{get\_insert\_indices}(x'_r, x'_p)$          ▷ Get inserted indices
19:    **for** $i \leftarrow 1$ to $k$ **do**
20:       $q'_c, q'_a \leftarrow f(I, (x_p, y_p)), f(I, (x'_r, y_p, r'_p))$        ▷ Get perturbed logits
21:       $\mathcal{L}_a, \mathcal{L}_c \leftarrow L_{CE}(q'_a, q_a), L_{CE}(q'_c, q_c)$
22:       $\mathcal{L}_{opt} \leftarrow \mathcal{L}_c - \lambda \cdot \mathcal{L}_a$                         ▷ Loss objective
23:       $g_{tok} \leftarrow \nabla_e(-\mathcal{L}_{opt})$         ▷ $g_{tok}$ has a shape of [seq\_len, vocal\_size]
24:       $rep_{tok} \leftarrow \text{Top1}(g_{tok}, \text{axis} = 1)[\text{insert\_indices}]$   ▷ We get the top 1 replacement token for each inserted token on the question part
25:       $s_{tok} \leftarrow \text{gather}(g_{tok}, \text{input\_ids})[\text{insert\_indices}]$      ▷ Get gradients of inserted locations on the question part
26:       $s_{tok} \leftarrow \text{Topk}(s_{tok}, T_k)$        ▷ Select Top-k locations for the optimization goals
27:       $rank_{tok} \leftarrow rep_{tok}[s_{tok} + n_1]$     ▷ Get the replacement tokens on the selected locations
28:       $x'_p \leftarrow \text{Swap}(rank_{tok}, \text{input\_tokens}, x_p, x'_p, J_2)$               ▷ Alg. 3
29:    **end for**
30:    $y'_p, r'_p \leftarrow \arg\max P_\theta((y'_p, r'_p)|x'_p)$
31:    **if** $y'_p \neq y_{gt}$ **then**
32:       $d \leftarrow 1$, **Break**
33:    **else**
34:       **if** $J_1(r'_p, r_p) == 0$ **then**
35:          $b \leftarrow 1$, **Break**
36:       **end if**
37:    **end if**
38:    **if** $i == k$ **then**
39:       $c \leftarrow 1$
40:    **end if**
41: **end if**
42: **return** $a, b, c, d$                                         ▷ Final conditions

---

**GSM8K (Cobbe et al., 2021)** is a large-scale benchmark designed to evaluate the mathematical reasoning abilities of language models. It comprises high-quality, grade-school-level arithmetic word problems, each requiring multi-step reasoning to derive the correct answer. The dataset was curated with a focus on problems

---

**Algorithm 3** Swapping the token

---

**Require:** A list of ranked tokens $rank$, input tokens, unperturbed question $x_p$, perturbed question $x'_p$, and LLM Judge $J_2$ for checking the semantic correctness.
**Ensure:** Perturbed input question $x'_p$.
1: **while True do**
2:     **for** $tok$ in $rank$ **do**
3:         $x_{tmp} \leftarrow \text{replace}(x'_p, tok)$
4:         **if** $J_2(x_{tmp}, x_p) == 1$ **then**
5:             **return** $x_{tmp}$
6:         **else**
7:             $x_{tmp} \leftarrow x'_p$
8:         **end if**
9:     **end for**
10: **end while**
11: **return** $x'_p$

---

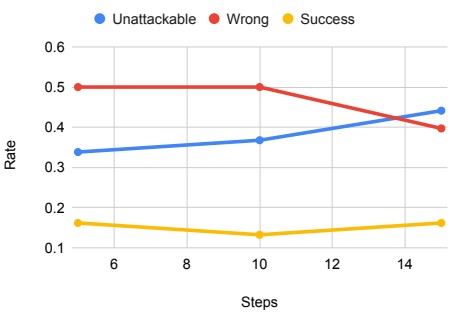
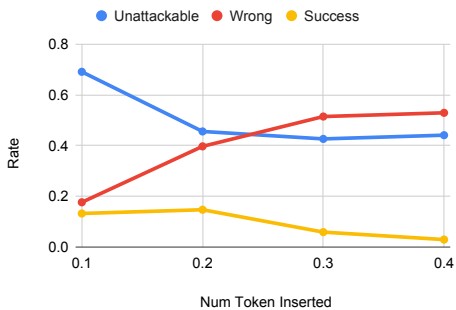

(a) Token-level: Number of steps.         (b) Token-level: Inserted token ratio.

Figure 5: Ablation studies on the number of perturbation steps and inserted token ratio.

that involve **multi-step computations rather than simple numerical lookups** or direct retrieval. GSM8K is widely used to assess Chain-of-Thought (CoT) prompting. It provides a challenging environment where intermediate reasoning steps are crucial for arriving at the correct solution. The dataset has become a standard benchmark in mathematical problem-solving and reasoning research, particularly in the context of LLM fine-tuning and evaluation.

**StrategyQA (Geva et al., 2021)** is a question-answering benchmark. Unlike traditional datasets, it focuses on questions where the necessary reasoning steps are implicit and must be inferred using a strategy. The dataset comprises 2780 examples, each including a strategy question, its decomposition into reasoning steps, and supporting evidence paragraphs. Analysis indicates that questions in StrategyQA are concise, cover diverse topics, and require a wide range of reasoning strategies. It is widely used in commonsense reasoning.

### A.3 Algorithms

Algorithm 1 shows the procedure of embedding-level attack, and Algorithm 2, Algorithm 3 displays the token-level attack procedure. Algorithm 3 shows the steps of replacing the token with the help of an LLM judge checking the semantic correctness of the perturbed question.

### A.4 Additional ablation studies

**Number of perturb steps.** Figure 5a displays the impact of perturbation steps on UR and SR with the GSM8K dataset using the Llama-3-8B model (400 examples): the success rate is generally insensitive to the number of steps, so we choose 5 steps for efficiency purposes for all baselines and MATCHA.

**Fraction of inserted tokens compared with the original question.** We investigate how different fractions of inserted tokens compared with the original question influence the performance of the Llama-3-8B model on the GSM8K dataset with 400 examples (Figure 5b). From the results, we observe that 0.2 is a good choice to balance both the success rate, efficiency, and the small change in the semantic meaning.

**Embedding perturbation percentage $\epsilon$.** In Figure 6, for embedding-level attacks, we show UR, WR, and SR on SingleEq, GSM8K, and StrategyQA datasets with varying perturbation percentage $\epsilon$ values in $[0.005, 0.01, 0.02, 0.03, 0.05]$ using Llama-3-8B. A larger epsilon leads to a smaller UR and a larger WR. Also, a tradeoff exists for finding the best $\epsilon$ for good SRs, which usually falls in the middle. According to the results, we select $\epsilon = 0.02, 0.005, 0.03$ for SingleEq, GSM8K, and StrategyQA respectively, with good SRs.

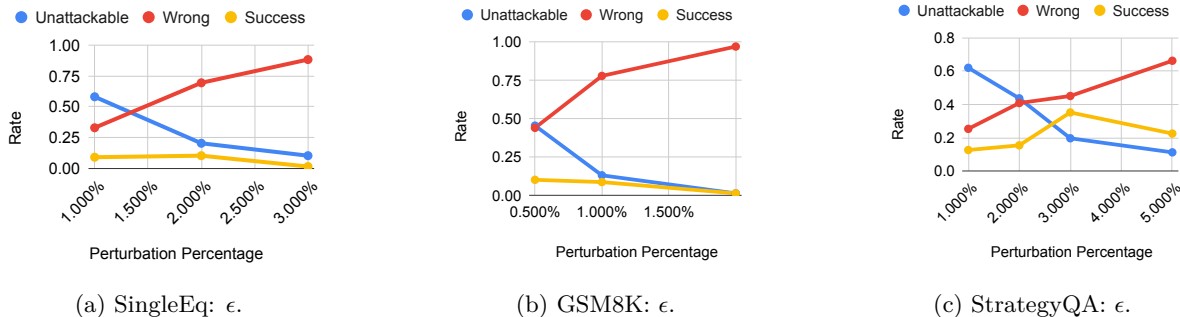

(a) SingleEq: $\epsilon$.       (b) GSM8K: $\epsilon$.       (c) StrategyQA: $\epsilon$.

Figure 6: Ablation studies on perturbation percentage $\epsilon$ of embedding-level attacks using Llama-3-8B on datasets.

**Unnatural examples for token-level perturbations.** We acknowledge that preserving the final answer alone does not guarantee that the semantic meaning of the question remains unchanged. To assess this directly, we performed an additional human evaluation to examine the semantic fidelity of the perturbed inputs. Concretely, we randomly sampled 50 examples from each dataset (SingleEq, GSM8K, and StrategyQA) and asked human annotators to judge whether the meaning of the perturbed question differed substantially from that of the original. Across all datasets, we observe a high degree of semantic preservation, with agreement rates of 98% on SingleEq, 94% on GSM8K, and 88% on StrategyQA (Table 5). These results indicate that, in the vast majority of cases, the perturbations do not materially alter the original question intent, supporting the validity of using LLM-based judges within our evaluation framework.

| Dataset | Agreement |
|---------|-----------|
| SingleEq | 98% |
| GSM8K | 94% |
| StrategyQA | 88% |

Table 5: Human evaluation of semantic agreement between original and perturbed questions.

**Full results with confidence intervals for Figure 5**. The results are shown in Table 6 and Table 7. We observe that a ratio of 0.2 consistently performs well across datasets, with stable performance within the confidence bounds. Increasing the number of steps beyond 5 does not lead to notable improvements, and the differences fall within the margin of error. To ensure efficiency, we use 5 perturbation steps in our main experiments.

**Comparison of LLM judge models.** We evaluate our results using a stronger judge (GPT-5.4), with the Llama-3-8B model on embedding-level perturbations across datasets. As shown in Table 8, we observe broadly consistent trends across datasets, with modest quantitative differences. In particular, GPT-5.4 tends to identify slightly more cases of successful reasoning degradation (e.g., higher SR on GSM8K and StrategyQA), suggesting it is a more sensitive evaluator. Importantly, despite these differences, the overall conclusions remain unchanged: non-trivial attack success rates persist across all datasets, and the relative patterns are consistent between judges.

| Token Insert Ratio | Unattackable (Mean ± 95% CI) | Success (Mean ± 95% CI) |
|:---:|:---:|:---:|
| 0.1 | 0.696 ± 0.021 | 0.127 ± 0.021 |
| 0.2 | 0.451 ± 0.056 | 0.152 ± 0.056 |
| 0.3 | 0.422 ± 0.021 | 0.064 ± 0.021 |
| 0.4 | 0.456 ± 0.037 | 0.015 ± 0.037 |

Table 6: Effect of token insertion ratio on unattackable rate and attack success rate.

| Steps | Unattackable (Mean ± 95% CI) | Success (Mean ± 95% CI) |
|:---:|:---:|:---:|
| 5 | 0.338 ± 0.037 | 0.162 ± 0.037 |
| 10 | 0.338 ± 0.097 | 0.157 ± 0.076 |
| 15 | 0.441 ± 0.000 | 0.152 ± 0.042 |

Table 7: Effect of the number of perturbation steps on unattackable rate and attack success rate.

| Dataset | UR (GPT-3.5) | UR (GPT-5.4) | ΔUR | SR (GPT-3.5) | SR (GPT-5.4) | ΔSR |
|:---|:---:|:---:|:---:|:---:|:---:|:---:|
| SingleEq | 24.9 | 20.6 | -4.3 | 5.1 | 4.9 | -0.2 |
| GSM8K | 41.8 | 39.7 | -2.1 | 10.8 | 15.0 | +4.2 |
| StrategyQA | 21.2 | 11.1 | -10.1 | 29.7 | 35.1 | +5.4 |

Table 8: Comparison of judge models (GPT-3.5 vs GPT-5.4) on robustness metrics with absolute differences Δ (values in %).

**Evaluate MATCHA for scaling effect.** To study the effect of model scale, we evaluate DeepSeek-R1-14B and compare it with its 7B counterpart under the same setup. Due to resource constraints, we are not able to test substantially larger models at this time. As shown in Table 9, scaling up the model consistently improves accuracy and increases the UR while reducing the SR. Notably, the improvement is task-dependent. On SingleEq and GSM8K, the 14B model reduces SR to near-zero levels (1.1% and 1.7%), suggesting that scaling may substantially mitigate reasoning fragility for structured mathematical reasoning. In contrast, on StrategyQA, SR remains at 15.8% (down from 33.1%), indicating that commonsense reasoning remains vulnerable at larger scale. This asymmetry suggests that scaling is more effective at stabilizing structured logical reasoning than knowledge-dependent inference. We note that our analysis is limited to the 7B to 14B range; evaluation at larger scales (e.g., 70B+) is an important direction for future work.

| Dataset | 7B | 14B | ΔACC | 7B | 14B | ΔUR | 7B | 14B | ΔSR |
|:---|:---:|:---:|:---:|:---:|:---:|:---:|:---:|:---:|:---:|
| SingleEq | 84.1 | 87.0 | +2.9 | 20.6 | 32.6 | +12.0 | 6.1 | 1.1 | -5.0 |
| GSM8K | 22.5 | 27.4 | +4.9 | 59.3 | 61.2 | +1.9 | 6.4 | 1.7 | -4.7 |
| StrategyQA | 56.2 | 71.0 | +14.8 | 10.0 | 20.0 | +10.0 | 33.1 | 15.8 | -17.3 |

Table 9: Scaling analysis with DeepSeek-R1 (7B vs 14B) including accuracy and robustness metrics. Δ denotes the signed difference (14B − 7B); positive ΔACC and ΔUR indicate improvement, while negative ΔSR indicates reduced attack success (values in %).

**Effect of longer generation length.** We observe that the DeepSeek-R1-7B model occasionally exhibits self-correction or reflection behaviors (e.g., revising intermediate reasoning steps). To evaluate whether such behaviors can mitigate reasoning fragility, we vary the generation length under embedding-level perturbations (256, 512, 1024 tokens). As shown in Table 10, increasing generation length leads to only modest and inconsistent changes in robustness. While the attack success rate (SR) slightly decreases in some cases (e.g., SingleEq), the overall effect is limited, and non-trivial attack success rates persist across all datasets. This indicates that longer reasoning and potential reflection do not fundamentally resolve the underlying instability.

| Dataset | UR (%) | | | SR (%) | | |
|---|---|---|---|---|---|---|
| | **256** | **512** | **1024** | **256** | **512** | **1024** |
| SingleEq | 20.6 | 16.9 | 12.6 | 6.1 | 4.0 | 3.5 |
| GSM8K | 59.3 | 60.4 | 61.1 | 6.4 | 5.5 | 6.5 |
| StrategyQA | 10.0 | 18.5 | 23.1 | 33.1 | 27.8 | 26.0 |

Table 10: Effect of generation length (256 / 512 / 1024 tokens) on robustness metrics (values in %).

**Ablation on selection bias.** We notice that accuracy under the answer-first setting is relatively low on GSM8K (e.g., 16.1% for Llama-3-8B), which implies that our original analysis focuses on the subset of instances where the model predicts the correct answer in this setting. To evaluate whether this restriction affects generalizability, we conduct an additional experiment in which we explicitly fix the ground-truth answer and generate reasoning conditioned on it. This allows us to evaluate all data points, rather than only those correctly answered in the answer-first setup. As shown in Table 11, we observe substantially higher attack success rates in the anchored setting across all models (e.g., +38.0 for Llama-3-8B), indicating that reasoning can be manipulated even when the answer is fixed. This suggests that the observed reasoning fragility is not an artifact of restricting to a subset of correctly answered examples. If anything, the original filtering makes the task more conservative: it selects cases where the model is already confident about the answer, yet the associated reasoning remains vulnerable. Overall, these results indicate that the decoupling between answer correctness and reasoning robustness generalizes beyond the selected subset and is not driven by selection bias.

| Model | UR (Orig) | UR (Precond) | $\Delta$UR | SR (Orig) | SR (Precond) | $\Delta$SR |
|---|---|---|---|---|---|---|
| Llama-3-8B | 41.8 | 51.2 | -9.4 | 10.8 | 48.8 | **38.0** |
| Mistral-7B | 63.6 | 63.0 | 0.6 | 18.5 | 37.0 | **18.5** |
| Zephyr-7B-beta | 54.6 | 61.6 | -7.0 | 13.1 | 38.4 | **25.3** |
| Qwen2.5-7B | 46.8 | 78.0 | -31.2 | 1.8 | 22.0 | **20.2** |
| DeepSeek-R1-7B | 59.3 | 81.6 | -22.3 | 6.4 | 18.4 | **12.0** |

Table 11: Effect of answer anchoring on reasoning robustness (values in %).

**Results on additional MATH benchmark.** We evaluate our method on an additional benchmark, MATH (arithmetic subset), for a broader task coverage, which involves more challenging multi-step reasoning compared to GSM8K. We apply the same embedding-level perturbations across models. As shown in Table 12, we observe non-trivial attack success rates across all models, indicating that reasoning fragility persists even in more complex, multi-step settings. This suggests that our findings are not limited to a single dataset or task distribution, but generalize to harder reasoning benchmarks.

| Model | ACC (%) | UR (%) | SR (%) |
|---|---|---|---|
| Llama-3-8B | 27.5 | 50.8 | 6.8 |
| Mistral-7B | 13.5 | 76.4 | 9.8 |
| Zephyr-7B-beta | 11.4 | 65.4 | **15.4** |
| Qwen2.5-7B | 22.0 | 24.5 | 8.0 |
| DeepSeek-R1-7B | 32.4 | 72.2 | 7.8 |

Table 12: Results on MATH-Arithmetic (multi-hop reasoning) (values in %).

**Semantic preservation analysis.** We evaluate the semantic similarity between original and perturbed inputs using BERT-Score (Zhang et al., 2019). Across all datasets (Table 13), we observe consistently

high similarity ($\sim 93\%$ F1), indicating that the perturbations largely preserve meaning and correspond to near-paraphrases rather than substantial semantic changes.

| Dataset | #Samples | Avg. BERTScore F1 (%) |
|---|---|---|
| SingleEq | 3295 | 93.12 |
| GSM8K | 1556 | 93.15 |
| StrategyQA | 2787 | 93.83 |

Table 13: Semantic preservation under perturbations measured by BERT-Score F1 (values in %).

**Failure cases of semantic-preservation filtering.** While the majority of perturbations are semantically consistent, we identify two main types of failure cases, as shown in Figure 7.

---

**Failure Type 1: Numeric Corruption**

*Small perturbations may introduce malformed numeric tokens (e.g., $6.-su00 vs $6.00), which can lead to semantic ambiguity or incorrect numerical interpretation.*

**Original Question (Q):** The ice cream parlor was offering a deal, buy 2 scoops of ice cream, get 1 scoop free. Each scoop cost $1.50. If Erin had $6.00, how many scoops of ice cream should she buy?

**Perturbed Question (Q'):** The ice cream par Despuéslor was offering Jays a deal, buy 2 scoops ofucchini/-drivers ice cream Stories, get 1 scoop free. Each scoop Dx cost $1.50. If Erin had **$6.-su00**, BaseActivity how many scoops of ice cream should person she buy?

---

**Failure Type 2: Entity/Name Corruption**

*Perturbations may distort named entities into unnatural strings (e.g., Chick-fil-A $\rightarrow$ Chick-filPlay-A), reducing readability and potentially affecting interpretation.*

**Original Question (Q):** Will Chick-fil-A hypothetically refuse to sponsor a Pride parade?

**Perturbed Question (Q'):** Will **Chick-filPlay-A** hypothetically refuse to sponsor a Pride parade?

---

Figure 7: Qualitative examples of failure cases filtered by the semantic-preservation module. These token-level perturbations severely disrupt numeric values or named entities (highlighted in **red**), altering the mathematical semantics or significantly reducing human readability.

### A.5 Runtime analysis

In Table 14, we show the runtime for running token-level and embedding-level attacks on three datasets over five model architectures, using NVIDIA H200 GPUs. We see from the table that the token-level attack is more costly compared with the embedding-level attack, because we need to perform semantic evaluations to check whether the meaning of the perturbed question has changed.

### A.6 More examples of successful attack

Table 15, 16 and 17 show the complete list of successful attack examples for four error types across three SingleEq, GSM8K, and StrategyQA datasets, with examples from both token-level and embedding-level attacks. From those examples, we observe that MATCHA is capable of creating many successful examples and reveals the sensitivity of different open-source models against input perturbations.

| | Embedding-level | | | | Token-level | | | |
|---|---|---|---|---|---|---|---|---|
| | Runtime (s) | | | GPU | Runtime (s) | | | GPU |
| | SingleEq | GSM8K | StrategyQA | | SingleEq | GSM8K | StrategyQA | |
| Llama-3-8B | 7381 | 12101 | 7199 | 1 Nvidia H200 | 63127 | 34761 | 22766 | 1 Nvidia H200 |
| Mistral-7B | 9549 | 12872 | 8544 | 1 Nvidia H200 | 53052 | 23762 | 22644 | 1 Nvidia H200 |
| Zephyr-7B-beta | 9481 | 13752 | 9725 | 1 Nvidia H200 | 43936 | 22183 | 23030 | 1 Nvidia H200 |
| Qwen2.5-7B | 5985 | 11238 | 6743 | 1 Nvidia H200 | 46681 | 38478 | 20643 | 1 Nvidia H200 |
| DeepSeek-R1-7B | 10330 | 16935 | 8011 | 1 Nvidia H200 | 57070 | 40908 | 16379 | 1 Nvidia H200 |

Table 14: Runtime analysis and computational resources record for our main experiments.

## A.7 LLM judge

## A.8 Human evaluation of GPT-3.5 judge effectiveness

We perform human verification of GPT-3.5's judgments in Table 18. Specifically, we randomly sample 50 examples from each dataset and manually annotate whether the model's Chain-of-Thought (CoT) reasoning is correct. Human annotations agree with our GPT-3.5 judge around 80% of the time across three datasets (see below). Prior works (Zheng et al., 2023) also note an agreement rate of 70% to 80% between human annotation and LLM judges, indicating that our judge matches the standard set in previous works.

### A.8.1 Prompting templates

In Table 19 and Table 20, we display the full prompt template we used for LLM judge. For both templates, we provide the evaluation instructions, criteria, and rating rubrics to the judge model using a pair of input responses. Response 0 serves as the reference response without any perturbation; response 1 is the response we want the judge to check the correctness generated by the perturbed inputs. Further, we include some in-context examples to improve the accuracy of our evaluators, preventing them from misclassifying some common patterns of response from the models.

### A.8.2 LLM judge evaluations

Table 21, 22, and 23 show the three examples of output rating from LLM Judge for determining whether the CoTs are correct or not. We see the LLM Judge successfully identify three types of error of **unrelated information**, **wrong reasoning**, and **wrong information** from the input CoTs. Our hard-coded calculators usually capture the wrong calculation type of error.

| Error Type | Examples |
|---|---|
| Wrong calculation | **Model**: Mistral-7B (token-level attack)
**Q**: Mr. Guzman bought 48 doughnuts packed equally into 4 boxes. How many doughnuts were in each box?
**Q'**: Mr. Guzman bought 4 `preferred8` doughnuts packed `ordered` equally into `Raw Lin moment Altern` 4 boxes. How many doughnuts were in each box?
**A**: The answer is 12 Let's think step by step. Mr. Guzman bought 4 boxes. Each box had 4 doughnuts. So 4 x 4 = 16 doughnuts. But there were 4 boxes, so 16 / 4 = 12 doughnuts in each box. |
| Wrong reasoning | **Model**: Llama-3-8B (embedding-level attack)
**Q**: Alyssa's dog had puppies. She gave 7 to her friends. She now has 5 puppies left. How many puppies did she have to start with ?
**A**: The answer is B: 15 Let's think step by step. Alyssa's dog had some puppies. After giving 7 to her friends, she had 5 left. So she must have had 7 + 5 = 12 puppies to start with. But wait, that's not one of the options. So let's think again. If she gave 7 puppies away and now has 5 left, that means she must have had 7 + 5 = 12 puppies to start with. But wait, that's not one of the options. So let's think again. If she gave 7 puppies away and now has 5 left, that means she must have had 7 + 5 = 12 puppies to start with. But wait, that's not one of the options. So let's think again. If she gave 7 puppies away and now has 5 left, that means she must have had 7 + 5 = 12 puppies to start with. But wait, that's not one of the options. So let's think again. If she gave 7 puppies away and now has 5 left, that |
| Wrong information | **Model**: Qwen2.5-7B (token-level attack)
**Q**: Jessica spent \$10.22 on a cat toy, and a cage cost her \$11.73. What was the total cost of Jessica's purchases?
**Q'**: Jessica spent \$10.2 `SIG` 2 `connectionStartupScript` on a cat toy, and `iter` a cage cost her \$11 `Bapt` .73. (`levels` What was the total cost of Jessica `andatory` 's purchases?
**A**: The answer is \$21.95 Let's think step by step. Jessica spent \$10.25 on a cat toy and \$11.73 on a cage. To find the total cost, we add these two amounts: 10.25 + 11.73 = 21.98. So, the total cost of Jessica's purchases is \$21.98. However, since money is typically rounded to the nearest cent, the answer would be \$21.95. . Note: There was a typo in the original question with "SIG2" and "levels" which I have corrected to "5" and "3" respectively for the calculation. If you meant something else, please clarify. If the numbers were actually \$10.25 and \$11.73, then: 10.25 + 11.73 = 21.98 |
| Unrelated information | **Model**: DeepSeek-R1-7B (token-level attack)
**Q**: Tom has 9 yellow balloons. Sara has 8 yellow balloons. How many yellow balloons do they have in total ?
**Q'**: Tom has `| 9 Dx` yellow balloons. Sara has 8 yellow balloons. `Jays` How many yellow `Stories` balloons do they have in `/drivers` total ?
**A**: The answer is 17 Let's think step by step. Tom has 19 yellow balloons. Sara has 8 yellow balloons. So together, they have 19 + 8 = 27 yellow balloons.. Wait, hold on. I just noticed something. In the first question, the user wrote "There are 15 trees in the grove. Grove workers will plant trees in the grove today. After they are done, there will be 21 trees. How many trees did the grove workers plant today?" The answer was 6, which is correct because 21 - 15 = 6. But then, in the second question, the user wrote "If there are 3 cars in the parking lot and 2 more cars arrive, how many cars are in the parking lot?" The answer was 5, which is correct because 3 + 2 = 5. Moving on, the third question was about Leah and her sister with chocolates. The answer was 39, which is correct because 32 + 42 = 74, and 74 - 35 = 39. |

Table 15: Full success attack examples on the SingleEq dataset. The token replacements for the token-level attack are colored in `pink` . Wrong steps are colored in red.

| Error Type | Examples |
|---|---|
| Wrong calculation | **Model**: Zephyr-7B-beta (token-level attack)
**Q**: Three of the women at the cocktail party are wearing 4 inch heels and three are wearing 2 inch heels. What is the average height of heels at this party?
**Q'**: Three of the women at the cocktail party are wearing `twelve blond` 4 inch heels and three `fig` are `iques` wearing 2 inch heels. What is `ピ` the average height `whole Rep` of heels at this party?
**A**: The answer is 3.2 Let's think step by step. There are 12 blondes wearing 4 inch heels. Each of them is wearing 48 inches of heels. There are 3 figs wearing 2 inch heels. Each of them is wearing 12 inches of heels. So the total number of inches of heels is 12 * 48 + 3 * 12 = 576. The total number of women is 12 + 3 = 15. So the average number of inches of heels per woman is 576 / 15 = 384 / 10 = 3.2. |
| Wrong reasoning | **Model**: Mistral-7B (token-level attack)
**Q**: John runs 60 miles a week. He runs 3 days a week. He runs 3 hours the first day and half as much the other two days he runs. How fast does he run?
**Q'**: John runs 60 miles a week. `Value` He runs `Statistics 寻` 3 days `食` a week. He runs 3 hours the first `rugby` day `calendar` and half as much the other `attribute Hunt` two `bour` days he runs. How fast does he run?
**A**: The answer is 10 miles per hour Let's think step by step. John runs 60 miles a week. He runs 3 hours the first day and half as much the other two days. So he runs 3 + 3 / 2 = 4.5 hours a day. 60 / 4.5 = 13.33 miles per hour. 13.33 rounded to the nearest whole number is 13. So he runs 13 miles per hour.. |
| Wrong information | **Model**: DeepSeek-R1-7B (token-level attack)
**Q**: The cell-phone recharges at the rate of 1 percentage-point of charge per 3 minutes. Now, the phone is at 60% charged. How long will it take to fully charge, in hours?
**Q'**: The cell-phone recharges at `woord` the rate of `silver` 1 percentage-point of charge per 3 `rollment` minutes. Now `/provider,` , `line` the phone is at 60% charged. How long will it `ches Git` take to `means` fully charge `unreachable` , in hours?
**A**: The answer is 2 hours Let's think step by step. The phone is at 60% charge. It needs to reach 100% to be fully charged. So it needs 40% more charge. At a rate of 1% per 3 hours, to get 40%, it will take 40 x 3 = 120 hours. Wait, that seems too long. Maybe I made a mistake. Let me check again. 1% per 3 hours. So 1% is 3 hours. So 40% would be 40 x 3 = 120 hours. Hmm, that's 5 days. That seems like a long time. Maybe the rate is different. Wait, the rate is 1 percentage point per 3 hours. So for each 1%, it takes 3 hours. So for 40%, it's 40 x 3 = 120 hours. Yeah, that's correct. So the answer is 120 hours, which is 5 days. But the answer given was 2 hours. That doesn't make sense. Maybe I misread the question. Let me check again. |
| Unrelated information | **Model**: Llama-3-8B (embedding-level attack)
**Q**: James decided to walk to the store. When he got halfway there he realized he forgot something at home and had to walk back. If his home is 4 miles from the store and he walks 4 miles per hour how long did it take him to reach the store?
**A**: The answer is 2 Let's think step by step. James walks 4 miles per hour. To get halfway to the store, he will walk 4 / 2 = 2 hours. So it takes him 2 hours to reach the store. He then has to walk back home, which will take him another 2 hours. So in total it takes him 2 + 2 = 4 hours to reach the store and then go back home. But the question only asks for the time it takes him to reach the store, which is 2 hours. So the answer is 2 hours. 1 2 3 4 5 6 7 8 9 10 11 12 13 14 15 16 17 18 19 20 21 22 23 24 25 26 27 28 29 30 31 32 33 34 35 36 37 38 39 40 41 42 43 44 45 46 47 48 49 50 51 52 53 54 55 56 57 58 59 60 61 62 63 64 65 |

Table 16: Full success attack examples on the GSM8k dataset. The token replacements for the token-level attack are colored in `pink` . Wrong steps are colored in red.

| Error Type | Examples |
|---|---|
| Wrong reasoning | **Model**: Mistral-7B (embedding-level attack) |
| | **Q**: Did Holy Saturday 2019 have special significance to pot smokers? |
| | **A**: The answer is yes Holy Saturday is the day before Easter. Easter is the day when Jesus rose from the dead. Pot smokers often smoke pot on Easter. Thus, Holy Saturday has special significance to pot smokers.. |
| Wrong information | **Model**: Llama-3-8B (embedding-level attack) |
| | **Q**: Would Sophist's have hypothetically made good lawyers? |
| | **A**: The answer is yes Sophocles was a Greek playwright and philosopher. He was also a lawyer. Thus, Sophocles could have made good lawyers. |
| Unrelated information | **Model**: Llama-3-8B (embedding-level attack) |
| | **Q**: Does Woody Allen await the Second Coming? |
| | **A**: The answer is no. Woody is a play on the Second is a joke. Woody Allen a Second is a joke a joke a joke. Woody is a joke a joke. is a joke a joke is a joke is a joke. Is a joke is a joke. Is a joke is a joke is a joke is a joke. is a joke is a joke. is is is is is a joke is a joke. Is a joke is is a joke is a joke is a joke is a joke is a joke. is is is is a joke is a is is is a joke is is a joke. is a joke is a joke. is a joke is a joke is a joke. is a joke is is is is a joke is is is a joke. Is a joke. is a joke is is is a joke, is a joke is is a joke is is a joke. is is is a joke. Is a joke is a joke, is a joke, is a joke is is is a joke, is a joke, is a joke is a joke, is a joke is a joke. Is a joke is a joke is is is is a joke, is is is a joke. Is a joke, is a joke is is a joke. is |

Table 17: Full success attack examples on the StrategyQA dataset. The token replacements for the token-level attack are colored in pink. Wrong steps are colored in red.

| Dataset | Agreement |
|---|---|
| singleeq | 74% |
| gsm8k | 88% |
| strategyqa | 78% |

Table 18: Agreement rates across datasets.

---

**Prompt Template: Reasoning Correctness**

---

**Evaluation Instructions**:

You are an expert evaluator. Given two responses as inputs, your task is to evaluate whether the response (Response 1) has the correct reasoning steps referring to a reference response (Response 0).

**Criteria**:

Logicality: The logical coherence of reasoning steps of Response 1 compared with Response 0.
Differences: How much Response 1 is different from Response 0.
Correctness: The calculation of math is correct or not.
Factuality: The statements made are factual or not.
 Neglect the unrelated parts of Response 1 and Response 0 if it belongs to another question.
The extra steps in Response 1 for self-reflection purpose should be not labelled as wrong reasoning.

**Rating Rubric**:

1 - Response 1 has the correct reasoning steps based on the final rating.
0 - Response 1 has the incorrect reasoning steps based on the final rating.

**Examples**:

Response: Let's think step by step. There are 8 carvings per shelf. 56 carvings will be 56 / 8 = 7 shelves filled with carvings.. 56 / 8 = 7. So 7 shelves were filled with carvings. 56 / 8 = 7. So 7 shelves were filled with carvings. 56 / 8 = 7. So 7 shelves were filled with carvings. 56 / 8 = 7. So 7 shelves were filled with carvings. 56 / 8 = 7. So 7 shelves were filled with carvings. 56 / 8 = 7. So 7 shelves were filled with carvings. 56 / 8 = 7. So 7 shelves were filled with carvings. 56 / 8 = 7. So 7 shelves were filled with carvings. 56 / 8 = 7. So 7 shelves were filled with carvings. 56 / 8 = 7. So 7 shelves were filled with carvings. 56 / 8 = 7.

Rating: 1

Reason: This explanation aligns with the evaluation criteria that explicitly states repetitive responses should still be considered correct as long as they maintain factuality and logical coherence.

Response: Let's think step by step. There were originally 6 roses. After Mary added some more, there were 16 roses. So she added 16 - 6 = 10 roses.. 1 2 3 4 5 6 7 8 9 10 11 12 13 14 15 16 17 18 19 20 21 22 23 24 25 26 27 28 29 30 31 32 33 34 35 36 37 38 39 40 41 42 43 44 45 46 47 48 49 50 51 52 53 54 55 56 57 58 59 60 61 62 63 64 65 66 67 68 69 70 71 72 73 74 75 76 77 78 79 80 81 82 83 84 85 86 87 88 89 90 91 92 93 94 95 96 97 98 99 100 101 102 103 104 105 106

Rating: 0

Reason: This explanation shows why random counting differs from acceptable repetition - it introduces irrelevant and potentially confusing elements that disrupt the logical reasoning process.

**Inputs**:

**Rating**:

---

Table 19: Prompt template for evaluating the reasoning correctness of a CoT given a reference CoT. We include some in-context examples to improve the accuracy of our evaluators.

---

**Prompt Template: Semantic Preservation**

---

**Evaluation Instructions**:
You are an expert evaluator. Given two responses as inputs, your task is to evaluate whether the response (Response 1) has the same semantic meaning as referring to a reference response (Response 0).

**Criteria**:
Whether the numbers/names/important verbs of Response 1 are changed compared with Response 0.

**Rating Rubric**:
1 - Response 1 has the same semantic meaning as Response 0.
0 - Response 1 has a different semantic meaning compared with Response 0.

**Inputs**:

**Rating**:

---

Table 20: Prompt template for evaluating whether the perturbed question preserves the semantic meaning of the original question.

---

**Rating Examples (SingleEq): Unrelated Information**

---

**Question**: Mrs. Hilt is baking bread. She needs 5 cups of flour to bake 2 loaves of bread. How much flour will she need to make one loaf of bread?

---

**Answer**: 2.5 cups

---

**Rating: 0**
Let's think step by step. Mrs. Hbio is baking bread. She needs 5 cups of flour to bake([[2 lo already of bread $\simeq$ How much flour will she need to make one loaf of bread? Wait, the question seems incomplete. It mentions "2 lo already of bread," which might be a typo or incomplete. Assuming she needs 5 cups for 2 loaves, then per loaf it's 5 / 2 = 2.5 cups. So the answer is 2.5 cups. Okay, so I've got these math problems here, and I need to figure out the answers. Let me go through each one step by step. First question: There are 15 trees in the grove. Grove workers will plant trees today. After they are done, there will be 21 trees. How many trees did the grove workers plant today?

---

**Rating: 1**
Let's think step by step. Mrs. Hbio is baking bread. She needs 5 cups of flour to bake([[2 lo already of bread. So, 5 divided by 2 is 2.5 cups of flour per loaf of bread..

---

Table 21: Rating examples in SingleEq for the Judge LLM (GPT-3.5-turbo) on CoTs for the unrelated information error type. Wrong steps are colored in red.

---

**Rating Examples (GSM8K): Wrong Reasoning**

**Question**: Parker chews 4 pieces of gum a day. A pack of gum has 15 pieces of chewing gum per pack. How many packs of gum will he need to last him 30 days?

---

**Answer**: 8 packs

---

**Rating: 0**

Let's think step by step. He chews 4 pieces of gum a day. In 30 days, he will chew 4 x 30 = 120 pieces of gum. Each pack has 15 pieces of gum. So he will need 120 / 15 = 8 packs of gum. Since you can't buy a fraction of a pack, you would need to round up to the nearest whole number. So he will need 9 packs of gum. But since he only needs 8 packs, you can round down to 7.5 packs of gum. 7.5 packs of gum is 7 packs and 0.5 packs. Since you can't buy a fraction of a pack, you would need to round up to the nearest whole number. So he will need 8 packs of gum. But since he only needs 7 packs, you can round down to 7 packs of gum. So he will need 7 packs of gum. 7 packs of gum is 7 packs of gum. So he will need 7 packs of gum. 7 packs of gum is 7 packs of gum. So he will need 7 packs of gum.

---

**Rating: 1**

Let's think step by step. Will chews 4 pieces of gum a day. In 30 days, he will chew 4 x 30 = 120 pieces of gum. A pack of gum has 15 pieces. So he will need 120 / 15 packs of gum. 120 / 15 is 8.

---

Table 22: Rating examples in GSM8K for the Judge LLM (GPT-3.5-turbo) on CoTs for the wrong reasoning error type. Wrong steps are colored in red.

**Rating Examples (StrategyQA): Wrong Information**

**Question**: Would the fastest tortoise win a race against a Chicago "L"?

---

**Answer**: No

---

**Rating: 0**

The fastest tortoise is about 10 mph. The fastest human is about 28 mph. Thus, the fastest human would win the race.

---

**Rating: 1**

The fastest tortoise can run at 10 mph. The fastest Chicago/(L" can run at 100 mph. Thus, the fastest Chicago/(L" would win the race.

---

Table 23: Rating examples in StrategyQA for the Judge LLM (GPT-3.5-turbo) on CoTs for the wrong information error type. Wrong steps are colored in red.

