# OpenReview forum: "Robust Answers, Fragile Logic: Probing the Decoupling Hypothesis in LLM Reasoning"
_TMLR — Accepted by TMLR_

### Review · Reviewer_Tmzw · 2026-02-24

**Summary Of Contributions:**

This paper investigates whether CoT reasoning in LLMs is causally tied to the final answer or serves as post-hoc rationalization (the "Decoupling Hypothesis"). The authors introduce MATCHA, a framework that uses an answer-conditioned probing setup: the model first commits to an answer, then generates reasoning. MATCHA constructs token-level (gradient-informed insertion/replacement) and embedding-level (PGD-style) perturbations optimized to disrupt CoT while preserving the answer. Experiments on five 7B models across three datasets show that small perturbations can cause inconsistent reasoning while the answer remains correct. The authors also show non-trivial transferability of adversarial examples to GPT-3.5-turbo and GPT-4o.

**Audience:**

Yes

**Audience Explanation:**

The question of whether LLM reasoning is faithful is important and timely. The MATCHA framework, viewed as a CoT robustness evaluation tool rather than a faithfulness probe, is technically well-executed. The error taxonomy (wrong calculations, wrong reasoning, wrong information, unrelated information) provides useful qualitative insight, and the finding that multi-step reasoning is more fragile than single-step aligns with intuitions about cascading errors. The transferability experiments are also informative.

**Broader Impact Concerns:**

No significant concerns

**Claims And Evidence:**

No

**Claims Explanation:**

The central claim that CoT reasoning is decoupled from the model's answer is not convincingly supported due to two core issues:

**The answer-first setup fundamentally alters the CoT mechanism**. The paper has the model generate the answer before the reasoning (Question → Answer → CoT). This is not how CoT works. The benefit of CoT is precisely that intermediate reasoning tokens autoregressively influence the final answer. By reversing this order, the paper removes this causal pathway and then concludes reasoning is not causally connected to the answer. This is circular — the setup induces post-hoc rationalization and then claims to detect it. The large accuracy drops (e.g., Llama-3-8B at 16.1% on GSM8K vs ~50-60% with standard CoT) confirm that standard CoT reasoning does substantial causal work, which actually undermines the decoupling hypothesis. Moreover, since MATCHA only evaluates questions answered correctly in this answer-first format, the analysis is restricted to a heavily selected subset of easy questions where the model can pattern-match the answer without reasoning.

**The paper conflates robustness with faithfulness**. These are distinct properties. Faithfulness: does the CoT reflect the model's actual internal computation? Robustness: does the CoT remain stable under perturbation? A model could genuinely reason via CoT but have that reasoning be sensitive to input noise — the perturbation shifts internal computation, and the CoT faithfully reflects the broken process. Establishing faithfulness requires causal interventions on internal representations, not input-level perturbations. The title and metrics implicitly acknowledge this is really about robustness, yet the framing throughout claims it's about faithfulness.
Critically, the paper lacks the most important baseline: applying MATCHA in the standard CoT setting (reasoning-first). Without this, we cannot tell whether the fragility is an artifact of the answer-first setup or a genuine property of model reasoning.

**Requested Changes:**

Major:
- Add a standard CoT baseline. Apply MATCHA perturbations in the reasoning-first setting. This is essential to determine whether the observed fragility is specific to the answer-first setup or generalizes to standard CoT usage.
- Reframe around robustness, not faithfulness. The experiments support claims about CoT robustness under perturbation, not about the causal relationship between reasoning and answers. Either reframe accordingly, or introduce experiments that can speak to faithfulness (e.g., causal tracing).
- Discuss the selection bias. On GSM8K, only 16.1% of questions are answered correctly in the answer-first format. Discuss how restricting analysis to this easy subset affects generalizability.

Minor:
- Consider using a stronger LLM judge than GPT-3.5-turbo (74-88% human agreement), especially given the modest success rates (5-15%) where measurement noise matters.
- Evaluate on larger models (70B+) where reasoning may be more robust.

---

> ### Author Response · Authors · 2026-03-25
> **Response**
>
> We thank the reviewer for their constructive feedback and provide detailed point-by-point responses below.
>
> **1. Add a standard CoT baseline.**
>
> We thank the reviewer for this important suggestion. We agree that evaluating standard CoT (CoT-then-Answer) prompting is necessary to determine whether the observed fragility is specific to the answer-conditioned setup. To address this, we implemented a CoT-then-Answer baseline with the same embedding-level perturbations. In this setting, it is impossible to explicitly fix or control the position of the generated final answer due to the autoregressive nature of LLMs. To approximate answer consistency, we identify the answer from the final portion of the generated sequence (e.g., the last quarter of tokens) and measure answer similarity based on this segment. Despite this constraint, we observe that reasoning instability persists in the standard CoT setting. Below are representative examples where small perturbations lead to clearly degraded or incoherent reasoning, even though the model attempts to follow a step-by-step format:
>
> - LLama3-8B:
>
> *Q: Carl has a cane that is half as long as he is tall.  Carl is one foot taller than his brother, Ned.  And Ned is two feet shorter than his cousin, Isabel.  If Isabel is 7 feet tall, how long is Carl's cane, in feet?*
>
> *A’: Let's think step by step. Isabel is://://.swing Isabel is the the.. the.  Carl is the taller than Ned by 7 feet.  Ned is 1 foot taller than Carl..  So Carl is 6 feet tall..  Carl's cane is half as long as he is tall..  So it is 3 feet long. The answer is 3.*
>
>
> - Zephyr-7B-beta:
>
> *Q: The vending machines sell chips for 40 cents and candy bars for 75 cents. George spent $5 and got 3 bags of chips and had 1% of his money left. How many candy bars did he buy?*
>
> *A’: Let's think step by step. George spent 5 dollars. 3 bags of chips cost 3 x 40 = 120 cents. So he spent 120 cents on chips. That leaves **5 - 120 = 380 cents** for candy bars and the remaining money. 1% of 5 dollars is 5 cents. So George has 380 - 5 = 375 cents left for candy bars. 375 cents is 5 candy bars at 75 cents each. So George bought 5 candy bars. The answer is 5.*
>
>
> Importantly, due to the autoregressive nature of LLMs, reasoning and answer generation are tightly coupled in the CoT-then-Answer setting. As a result, when the reasoning chain diverges, the final answer often becomes incorrect simultaneously. This makes it difficult to isolate reasoning-specific failures using answer correctness alone. To account for this, we adapt our attack success criterion in the CoT-then-Answer setting to focus on reasoning quality, evaluated using the same GPT-based judge. While this provides a practical way to assess reasoning instability under standard CoT prompting, we acknowledge that developing more precise methods to disentangle reasoning and answer generation in this setting remains an important direction for future work. Under this criterion, we observe non-trivial attack success rates across all models on GSM8K, as shown below:
>
>
> |        Model      |   ACC (C→A)  |   ACC (A→C)  |   UR (C→A)  |   UR (A→C)  |   SR (C→A)  |   SR (A→C)  |
> |:-----------------:|:------------:|:------------:|:-----------:|:-----------:|:-----------:|:-----------:|
> |   Llama-3-8B      | **75.6**  |   16.1       | **55.3** |   41.8      | **17.5** |   10.8      |
> |   Mistral-7B      | **45.4**  |   11.4       | **63.9** |   63.6      | **17.2** |   18.5      |
> |   Zephyr-7B-beta  | **43.8**  |   9.9        |   42.9      | **54.6** | **24.7** |   13.1      |
> |   Qwen2.5-7B      | **60.4**  |   24.8       | **60.3** |   46.8      | **6.3**  |   1.8       |
> |   DeepSeek-R1-7B  | **85.2**  |   22.5       | **80.8** |   59.3      | **7.7**  |   6.4       |

---

> > ### Author Response · Authors · 2026-03-25
> > **Response (2)**
> >
> > **2. Selection bias.**
> >
> > We thank the reviewer for raising this important concern regarding potential selection bias. We acknowledge that accuracy under the answer-first setting is relatively low on GSM8K (e.g., 16.1% for Llama-3-8B), which implies that our original analysis focuses on the subset of instances where the model predicts the correct answer in this setting.
> >
> > To evaluate whether this restriction affects generalizability, we conduct an additional experiment in which we explicitly fix the ground-truth answer and generate reasoning conditioned on it. This allows us to evaluate all data points, rather than only those correctly answered in the answer-first setup.
> >
> > As shown in the table below, we observe substantially higher attack success rates in the anchored setting across all models (e.g., +38.0 for Llama-3-8B), indicating that reasoning can be manipulated even when the answer is fixed. This suggests that the observed reasoning fragility is not an artifact of restricting to a subset of correctly answered examples. If anything, the original filtering makes the task more conservative: it selects cases where the model is already confident about the answer, yet the associated reasoning remains vulnerable.
> >
> > Overall, these results indicate that the decoupling between answer correctness and reasoning robustness generalizes beyond the selected subset and is not driven by selection bias.
> >
> > | Model          | UR (Orig) | UR (Precond) | ΔUR   | SR (Orig) | SR (Precond) | ΔSR      |
> > |----------------|-----------|--------------|-------|-----------|--------------|----------|
> > | Llama-3-8B     | 41.8      | 51.2         | -9.4  | 10.8      | 48.8         | **38.0** |
> > | Mistral-7B     | 63.6      | 63.0         | 0.6   | 18.5      | 37.0         | **18.5** |
> > | Zephyr-7B-beta | 54.6      | 61.6         | -7.0  | 13.1      | 38.4         | **25.3** |
> > | Qwen2.5-7B     | 46.8      | 78.0         | -31.2 | 1.8       | 22.0         | **20.2** |
> > | DeepSeek-R1-7B | 59.3      | 81.6         | -22.3 | 6.4       | 18.4         | **12.0** |
> >
> > **3. Reframe around robustness, not faithfulness.**
> >
> > Thanks for this great suggestion. We have updated the manuscript by reframing it around robustness.
> >
> > **4. Using a stronger LLM judge.**
> >
> > We follow the reviewer’s suggestion and evaluate our results using a stronger judge (GPT-5.4), with the Llama-3-8B model on embedding-level perturbations. As shown in the table below, we observe broadly consistent trends across datasets, with modest quantitative differences. In particular, GPT-5.4 tends to identify slightly more cases of successful reasoning degradation (e.g., higher SR on GSM8K and StrategyQA), suggesting it is a more sensitive evaluator. Importantly, despite these differences, the overall conclusions remain unchanged: non-trivial attack success rates persist across all datasets, and the relative patterns are consistent between judges.
> >
> > | Dataset    | UR (GPT-3.5) | UR (GPT-5.4) | ΔUR   | SR (GPT-3.5) | SR (GPT-5.4) | ΔSR  |
> > |------------|--------------|--------------|-------|--------------|--------------|------|
> > | SingleEq   | 24.9         | 20.6         | -4.3  | 5.1          | 4.9          | -0.2 |
> > | GSM8K      | 41.8         | 39.7         | -2.1  | 10.8         | 15.0         | +4.2 |
> > | StrategyQA | 21.2         | 11.1         | -10.1 | 29.7         | 35.1         | +5.4 |
> >
> > **5. Evaluate on larger models.**
> >
> > To study the effect of model scale, we evaluate DeepSeek-R1-14B and compare it with its 7B counterpart under the same setup. Due to resource constraints, we are not able to test substantially larger models at this time. As shown in the table below, scaling up the model consistently improves accuracy and increases the UR while reducing the SR. However, despite these improvements, non-trivial attack success rates persist across all datasets, indicating that reasoning fragility is not eliminated by scaling. Overall, these results suggest that while larger models are more robust, the underlying decoupling between answer correctness and reasoning robustness remains.
> >
> >
> > |     Dataset   |   ACC (7B)  |   ACC (14B)  |   Δ ACC  |   UR (7B)  |   UR (14B)  |   Δ UR  |   SR (7B)  |   SR (14B)  |   Δ SR  |
> > |:-------------:|:-----------:|:------------:|:--------:|:----------:|:-----------:|:-------:|:----------:|:-----------:|:-------:|
> > |   SingleEq    |   84.1      |   87.0       |   2.9    |   20.6     |   32.6      |   -12.0  |   6.1      |   1.1       |   5.0   |
> > |   GSM8K       |   22.5      |   27.4       |   4.9    |   59.3     |   61.2      |   -1.9   |   6.4      |   1.7       |   4.7   |
> > |   StrategyQA  |   56.2      |   71.0       |   14.8   |   10.0     |   20.0      |  -10.0  |   33.1     |   15.8      |   17.3  |

---

### Review · Reviewer_E7T9 · 2026-02-25

**Summary Of Contributions:**

I found this paper to be a thoughtful and fairly thorough investigation of a phenomenon many of us have noticed anecdotally: LLMs can often give the right final answer, but the accompanying chain-of-thought can be fragile, inconsistent, or even outright wrong. The authors formalize this intuition via the *Decoupling Hypothesis* and propose an “answer-conditioned” probing setup where the model essentially commits to an answer first and then generates a rationale conditioned on that answer. In my view, this is a nice way to stress-test whether the reasoning is actually robust or more like a post-hoc explanation.

Methodologically, the main contribution is **MATCHA**, which produces small adversarial perturbations that aim to keep the final answer correct while breaking the rationale. The paper explores both token-level perturbations (random insertion + gradient-based replacement with a semantic-preservation check) and embedding-level perturbations (bounded ( \ell_\infty ) attacks in embedding space). The authors evaluate on SingleEq, GSM8K, and StrategyQA across multiple open-source models, and they also show some transfer of token-level adversarial examples to closed-source GPT models. Overall, the experiments support the story that “robust answers” and “robust logic” can come apart quite dramatically.

**Strengths I see**

* I like the framing and setup: separating “answer correctness” from “rationale correctness” and probing them independently feels timely and useful.
* MATCHA is reasonably clear and well-specified; the objectives and attack variants are understandable, and the paper includes baselines and ablations.
* The empirical sweep across datasets and multiple models, plus transfer to GPT models, makes the results feel less like a one-off artifact.

**Weaknesses / limitations**

* The biggest concern for me is the reliance on an LLM judge to decide whether a rationale is “correct.” The paper reports decent but imperfect agreement with humans, but since this is central to UR/SR, I’d want stronger validation here.
* The answer-conditioned protocol is clever, but it’s also a different generation process than typical CoT usage. I think the paper would benefit from being more explicit about what portion of the effect is “true decoupling” vs “decoupling induced by the answer-first setup.”
* The token-level attacks sometimes look a bit unnatural (even if semantics are allegedly preserved). I’d like a clearer characterization of how “realistic” these perturbations are across tasks.

**Additional Comments:**

I enjoyed reading this overall, and I think the authors did a good job presenting a clean empirical story with reasonable ablations and baselines. The paper feels timely and relevant, and I especially liked the attempt to separate answer robustness from rationale robustness in a way that can be measured.

My main suggestion is: if the authors can strengthen the “reasoning correctness” evaluation (more human validation / better automatic checks) and more clearly separate “decoupling” from “protocol-induced” effects, the paper would become substantially more compelling

**Audience:**

Yes

**Audience Explanation:**

I think a meaningful slice of TMLR’s audience would care about this. Anyone working on LLM reasoning, robustness, explanation faithfulness, or evaluation methodology will recognize the practical issue: we’re increasingly treating CoT as if it provides insight into the model’s reasoning, yet it may be fragile or post-hoc. This paper offers a fairly direct way to probe that gap and provides empirical evidence that the gap is substantial on common reasoning benchmarks.

**Broader Impact Concerns:**

I don’t see major broader-impact red flags, but there is a clear dual-use angle: MATCHA could be used to deliberately produce misleading rationales while keeping answers correct, which could be harmful in settings where people trust explanations (education, medical advice, policy support, etc.). I’d like the authors to explicitly acknowledge this risk and frame MATCHA as primarily an auditing/evaluation tool, with some discussion of mitigations (e.g., discouraging reliance on free-form CoT as a guarantee of faithfulness; exploring more verifiable intermediate artifacts).

**Claims And Evidence:**

Yes

**Claims Explanation:**

Overall I think the authors do a good job backing up their main claims with evidence. They introduce a concrete framework (answer-conditioned probing + MATCHA), report consistent quantitative results across multiple datasets and several models, and include comparisons to simpler baselines and useful ablations (e.g., varying insert ratio/steps/(\epsilon)). The transferability experiments to GPT models also help convince me this isn’t purely a white-box artifact.

That said, I don’t think the evidence is *perfectly* conclusive because the key “reasoning correctness” signal is mostly produced by an LLM judge, and the reported human agreement is not 100% (and varies by dataset). I’m comfortable saying the *directional* conclusion is well-supported (rationales are much easier to break than answers), but I’d like the exact UR/SR numbers to be corroborated more strongly with human evaluation or task-specific checkers.

**Requested Changes:**

I’ll separate these into things I personally view as **critical** (would strongly affect my recommendation) vs changes that would **strengthen** the paper.

1. **[Critical] Clarify what is due to “decoupling” vs what is due to answer-conditioned generation.**
   The answer-first protocol is interesting, but it’s also nonstandard. I’d like extra experiments that compare:

   * standard CoT prompting vs
   * answer-conditioned rationale generation,
     ideally controlling for cases where the final answer is the same. This would help isolate whether we’re measuring a general property of the model or something amplified by the protocol.

2. **[Strengthening] Additional validation of “wrong reasoning” beyond LLM-as-a-judge.**
The LLM judge appears reasonably aligned with humans, but since it drives the main metrics, I’d appreciate a small human audit and/or automatic verification on arithmetic subsets. This would help quantify judge noise and provide confidence that UR/SR trends remain under alternative evaluators.

3. **[Strengthening] Better characterization of perturbation naturalness and semantic preservation.**
   For token-level attacks in particular, I’d like additional analysis/metrics on fluency/naturalness and failure cases for the semantic-preservation filter (and possibly stricter filtering to keep attacks “realistic”).

4. **[Strengthening] Broader task/model coverage (if feasible).**
   Adding one more benchmark (e.g., multi-hop QA or symbolic reasoning) and/or a larger model size would help make the conclusions feel more general and show whether scaling changes the robustness gap.

5. **[Strengthening] Statistical reporting and reproducibility details.**
   Confidence intervals / standard errors for UR/SR and transfer rates would help, and clarity about random seeds and exemplar choice effects for ICL would improve reproducibility.

6. **[Strengthening] Make the threat model more explicit.**
   In particular: what “imperceptible” means for token-level perturbations, and how to interpret embedding ( \ell_\infty ) bounds in terms of actual tokenization stability / plausible user inputs.

---

> ### Author Response · Authors · 2026-03-25
> **Response**
>
> We thank the reviewer for their constructive feedback and provide detailed point-by-point responses below. Also, we incorporated them in the updated version of the manuscript.
>
> **1. Standard CoT prompting baseline.**
>
> We thank the reviewer for this important suggestion. We agree that evaluating standard CoT (CoT-then-Answer) prompting is necessary to determine whether the observed fragility is specific to the answer-conditioned setup. To address this, we implemented a CoT-then-Answer baseline with the same embedding-level perturbations. In this setting, it is impossible to explicitly fix or control the position of the generated final answer due to the autoregressive nature of LLMs. To approximate answer consistency, we identify the answer from the final portion of the generated sequence (e.g., the last quarter of tokens) and measure answer similarity based on this segment. Despite this constraint, we observe that reasoning instability persists in the standard CoT setting. Below are representative examples where small perturbations lead to clearly degraded or incoherent reasoning, even though the model attempts to follow a step-by-step format:
>
> - LLama3-8B:
>
> *Q: Carl has a cane that is half as long as he is tall.  Carl is one foot taller than his brother, Ned.  And Ned is two feet shorter than his cousin, Isabel.  If Isabel is 7 feet tall, how long is Carl's cane, in feet?*
>
> *A’: Let's think step by step. Isabel is://://.swing Isabel is the the.. the.  Carl is the taller than Ned by 7 feet.  Ned is 1 foot taller than Carl..  So Carl is 6 feet tall..  Carl's cane is half as long as he is tall..  So it is 3 feet long. The answer is 3.*
>
>
> - Zephyr-7B-beta:
>
> *Q: The vending machines sell chips for 40 cents and candy bars for 75 cents. George spent $5 and got 3 bags of chips and had 1% of his money left. How many candy bars did he buy?*
>
> *A’: Let's think step by step. George spent 5 dollars. 3 bags of chips cost 3 x 40 = 120 cents. So he spent 120 cents on chips. That leaves **5 - 120 = 380 cents** for candy bars and the remaining money. 1% of 5 dollars is 5 cents. So George has 380 - 5 = 375 cents left for candy bars. 375 cents is 5 candy bars at 75 cents each. So George bought 5 candy bars. The answer is 5.*
>
>
> Importantly, due to the autoregressive nature of LLMs, reasoning and answer generation are tightly coupled in the CoT-then-Answer setting. As a result, when the reasoning chain diverges, the final answer often becomes incorrect simultaneously. This makes it difficult to isolate reasoning-specific failures using answer correctness alone. To account for this, we adapt our attack success criterion in the CoT-then-Answer setting to focus on reasoning quality, evaluated using the same GPT-based judge. While this provides a practical way to assess reasoning instability under standard CoT prompting, we acknowledge that developing more precise methods to disentangle reasoning and answer generation in this setting remains an important direction for future work. Under this criterion, we observe non-trivial attack success rates across all models on GSM8K, as shown below:
>
>
> |        Model      |   ACC (C→A)  |   ACC (A→C)  |   UR (C→A)  |   UR (A→C)  |   SR (C→A)  |   SR (A→C)  |
> |:-----------------:|:------------:|:------------:|:-----------:|:-----------:|:-----------:|:-----------:|
> |   Llama-3-8B      | **75.6**  |   16.1       | **55.3** |   41.8      | **17.5** |   10.8      |
> |   Mistral-7B      | **45.4**  |   11.4       | **63.9** |   63.6      | **17.2** |   18.5      |
> |   Zephyr-7B-beta  | **43.8**  |   9.9        |   42.9      | **54.6** | **24.7** |   13.1      |
> |   Qwen2.5-7B      | **60.4**  |   24.8       | **60.3** |   46.8      | **6.3**  |   1.8       |
> |   DeepSeek-R1-7B  | **85.2**  |   22.5       | **80.8** |   59.3      | **7.7**  |   6.4       |

---

> > ### Author Response · Authors · 2026-03-25
> > **Response (2)**
> >
> > **2. Additional validation of “wrong reasoning” beyond LLM-as-a-judge.**
> >
> > - Stronger LLM judge:
> >
> > We evaluate our results using a stronger judge (GPT-5.4), with the Llama-3-8B model on embedding-level perturbations across datasets. As shown in the table below, we observe broadly consistent trends across datasets, with modest quantitative differences. In particular, GPT-5.4 tends to identify slightly more cases of successful reasoning degradation (e.g., higher SR on GSM8K and StrategyQA), suggesting it is a more sensitive evaluator. Importantly, despite these differences, the overall conclusions remain unchanged: non-trivial attack success rates persist across all datasets, and the relative patterns are consistent between judges.
> >
> > | Dataset    | UR (GPT-3.5) | UR (GPT-5.4) | ΔUR   | SR (GPT-3.5) | SR (GPT-5.4) | ΔSR  |
> > |------------|--------------|--------------|-------|--------------|--------------|------|
> > | SingleEq   | 24.9         | 20.6         | -4.3  | 5.1          | 4.9          | -0.2 |
> > | GSM8K      | 41.8         | 39.7         | -2.1  | 10.8         | 15.0         | +4.2 |
> > | StrategyQA | 21.2         | 11.1         | -10.1 | 29.7         | 35.1         | +5.4 |
> >
> > - Automatic verification:
> >
> > For arithmetic tasks, we incorporate rule-based verification within the MATCHA framework (e.g., regex-based checking of intermediate calculations), which provides an independent, non-LLM signal for detecting incorrect reasoning. We observe consistent trends under this evaluation as well.
> >
> > Taken together, these results indicate that our findings are robust across multiple evaluators (stronger LLM judge, human annotation, and task-specific verification), and are unlikely to be artifacts of a particular judging mechanism.
> >
> > **3. Naturalness and semantic preservation.**
> >
> > We thank the reviewer for this suggestion and provide additional analysis on the realism of our perturbations.
> >
> > - Semantic preservation metrics.
> >
> > We evaluate the semantic similarity between original and perturbed inputs using BERTScore. Across all datasets, we observe consistently high similarity (≈93% F1), indicating that the perturbations largely preserve meaning and correspond to near-paraphrases rather than substantial semantic changes.
> >
> > |     Dataset   |   #Samples  |   Avg. BERTScore F1  |
> > |:-------------:|:-----------:|:--------------------:|
> > |   SingleEq    |   3295      |   93.12              |
> > |   GSM8K       |   1556      |   93.15              |
> > |   StrategyQA  |   2787      |   93.83              |
> >
> > - Failure cases of semantic-preservation filtering.
> >
> > While the majority of perturbations are semantically consistent, we identify two main types of failure cases:
> >
> > **Type 1: Numeric corruption**. Small perturbations may introduce malformed numeric tokens (e.g., $6.-su00 vs $6.00), which can lead to semantic ambiguity or incorrect numerical interpretation.
> >
> > *Q: The ice cream parlor was offering a deal, buy 2 scoops of ice cream, get 1 scoop free.  Each scoop cost $1.50.  If Erin had  $6.00, how many scoops of ice cream should she buy?*
> >
> > *Q’: The ice cream par Despuéslor was offering Jays a deal,丨 buy 2 scoops ofucchini/drivers ice cream Stories, get 1 scoop free. יות Each scoop Dx cost $1.50.  If Erin had  **$6.-su00**, BaseActivity how many scoops of ice cream should person she buy?*
> >
> > **Type 2: Entity/name corruption.** Perturbations may distort named entities into unnatural strings (e.g., Chick-fil-A → Chickเจร-f，一ilPlay-A), reducing readability and potentially affecting interpretation.
> >
> > *Q: Will Chick-fil-A hypothetically refuse to sponsor a Pride parade?*
> >
> > *Q’: Will **Chickเจร-f，一ilPlay-A** hypothetically refuse to sponsor a Pride parade?*

---

> > > ### Author Response · Authors · 2026-03-25
> > > **Response (3)**
> > >
> > > **4. Broader task/model coverage.**
> > >
> > > - Broader task coverage.
> > >
> > > We follow the reviewer’s suggestion and evaluate our method on an additional benchmark, MATH (arithmetic subset), which involves more challenging multi-step reasoning compared to GSM8K. We apply the same embedding-level perturbations across models. As shown in the table below, we observe non-trivial attack success rates across all models, indicating that reasoning fragility persists even in more complex, multi-step settings. This suggests that our findings are not limited to a single dataset or task distribution, but generalize to harder reasoning benchmarks.
> > >
> > > | Model          | ACC (%) | UR (%) | SR (%) |
> > > |----------------|---------|--------|--------|
> > > | Llama-3-8B     | 27.5    | 50.8   | 6.8    |
> > > | Mistral-7B     | 13.5    | 76.4   | 9.8    |
> > > | Zephyr-7B-beta | 11.4    | 65.4   | 15.4   |
> > > | Qwen2.5-7B     | 22.0    | 24.5   | 8.0    |
> > > | DeepSeek-R1-7B | 32.4    | 72.2   | 7.8    |
> > >
> > >
> > > - A larger model size.
> > >
> > > To study the effect of model scale, we evaluate DeepSeek-R1-14B and compare it with its 7B counterpart under the same setup. Due to resource constraints, we are not able to test substantially larger models at this time. As shown in the table below, scaling up the model consistently improves accuracy and increases the UR while reducing the SR. However, despite these improvements, non-trivial attack success rates persist across all datasets, indicating that reasoning fragility is not eliminated by scaling. Overall, these results suggest that while larger models are more robust, the underlying decoupling between answer correctness and reasoning robustness remains.
> > >
> > >
> > > |     Dataset   |   ACC (7B)  |   ACC (14B)  |   Δ ACC  |   UR (7B)  |   UR (14B)  |   Δ UR  |   SR (7B)  |   SR (14B)  |   Δ SR  |
> > > |:-------------:|:-----------:|:------------:|:--------:|:----------:|:-----------:|:-------:|:----------:|:-----------:|:-------:|
> > > |   SingleEq    |   84.1      |   87.0       |   2.9    |   20.6     |   32.6      |   -12.0  |   6.1      |   1.1       |   5.0   |
> > > |   GSM8K       |   22.5      |   27.4       |   4.9    |   59.3     |   61.2      |   -1.9   |   6.4      |   1.7       |   4.7   |
> > > |   StrategyQA  |   56.2      |   71.0       |   14.8   |   10.0     |   20.0      |  -10.0  |   33.1     |   15.8      |   17.3  |
> > >
> > >
> > > **5. Reproducibility.**
> > >
> > > **Random seed**. All experiments are conducted with a fixed random seed (seed = 1), ensuring consistency across runs.
> > >
> > > **Exemplar choice**. For in-context learning (ICL), we use standardized exemplars from the public repository: https://github.com/shizhediao/active-prompt/tree/main/basic_cot_prompts. Specifically, we use math_word_problems for math-related benchmarks (e.g., GSM8K, MATH) and strategyqa_problems for commonsense reasoning benchmarks (e.g., StrategyQA). This ensures that our results are based on established prompting templates and can be reproduced by prior work.
> > >
> > > **6. Threat model clarification.**
> > >
> > > For token-level perturbations, we define “imperceptible” as modifications that preserve semantic meaning and remain natural to human readers. Empirically, we verify this using BERTScore, which shows high semantic similarity (≈93% across datasets), indicating that the perturbed inputs are close paraphrases rather than meaning-altering edits.
> > >
> > > For embedding-level perturbations, we constrain changes within a small $\ell_\infty$ norm ball in the embedding space. In practice, these perturbations do not alter the tokenization or discrete input tokens, and therefore correspond to subtle variations within the model’s internal representation rather than unrealistic or out-of-distribution inputs.
> > >
> > > We revised our manuscript to reflect this clarification.
> > >
> > > **7. Broader impact concerns.**
> > >
> > > Thanks for pointing out the potential concerns. We have added a broader impact section in the updated paper.

---

> > > > ### Comment · Reviewer_E7T9 · 2026-03-26
> > > > **Response to authors**
> > > >
> > > > Thanks for the revision, I think the paper is substantially improved. In particular, the added human validation of the judge (Table 18), the semantic-preservation checks for perturbations (Table 5 / Table 13), the additional benchmark and scaling results (Table 12 / Table 9), and the added broader impacts section address the main concerns I raised.
> > > >
> > > > One minor suggestion though! In the abstract/introduction, I’d recommend slightly tightening the language around the CoT→Answer baseline. Section 5.5 is appropriately careful that standard CoT does not allow “answer-fixed” isolation due to autoregression and requires an adapted criterion; it would be good if the top-level claim (“not an artifact”) cannot be read as strict equivalence between the two settings. With that clarification, I’m satisfied that the requested changes have been addressed! Great job.

---

> > > > > ### Author Response · Authors · 2026-03-28
> > > > > **Reply**
> > > > >
> > > > > Dear Reviewer E7T9,
> > > > >
> > > > > Thank you for the thoughtful feedback and for taking the time to review our revisions. We’re glad that the additional experiments and analyses address your main concerns.
> > > > >
> > > > > We also appreciate your suggestion regarding the CoT→Answer baseline. We have refined the framing accordingly and uploaded an updated revision, with the revised statement below:
> > > > >
> > > > > *"We show that this fragility is not solely an artifact of our answer-conditioned protocol: while standard CoT-then-Answer generation does not permit strict answer-fixed isolation, it nevertheless exhibits similar patterns of reasoning degradation under analogous attacks."*
> > > > >
> > > > > Thanks again for the constructive feedback, and please feel free to let us know if you have any further comments.
> > > > >
> > > > > Best,
> > > > >
> > > > > Authors

---

### Review · Reviewer_sXF2 · 2026-03-10

**Summary Of Contributions:**

The paper investigates the "Decoupling Hypothesis" in LLMs, proposing that the generation of intermediate reasoning steps and the final answer prediction are not strictly causally linked, but rather loosely correlated. To test this, the authors introduce MATCHA. By forcing the model to generate the answer before generating CoT, MATCHA isolates the reasoning phase and tests it without altering the correct answer. The authors evaluate multiple open-source models and demonstrate that imperceptible perturbations can cause models to generate wrong reasoning steps for correct answers. They also show these adversarial examples transfer to closed-source models like GPT-4o.

Strengths:

1.The framework provides a novel perspective on evaluating the faithfulness and robustness of LLM explanations by explicitly targeting the reasoning-answer consistency.

2.Comprehensive evaluation: The approach of both token-level and embedding-level perturbations offers a stress test across varied architectures of models.

Weakness:

1.Paradigm mismatch: The paper evaluates an "answer-first, then reasoning" paradigm to isolate failures in reasoning process. However, the standard and widely adopted CoT paradigm relies on generating the reasoning first to drive the final answer. This structural mismatch undermines the paper's broad claim that standard CoT is inherently fragile, as the experiments actually test post-hoc rationalization rather than true step-by-step reasoning.

**Audience:**

Yes

**Audience Explanation:**

The concept of LLMs being "right for the wrong reasons" exposes a vulnerability in how we currently trust model outputs. Researchers focusing on interpretability and alignment of large language models, AI safety and adversarial attacks will find the MATCHA framework and the decoupling phenomenon relevant to their work.

**Broader Impact Concerns:**

While this work poses no significant immediate risks, in high-stakes domains, malicious actors could add imperceptible perturbations to generate false but convincing post-hoc explanations using this framework. This could mislead users who lack the specific domain knowledge to verify the flawed reasoning.

**Claims And Evidence:**

No

**Claims Explanation:**

1.While the authors provide extensive empirical results showing that models can generate inconsistent text when forced to rationalize a fixed answer, this evidence does not convincingly support the broad claim that standard CoT is fragile. The method relies on an "answer-first, then reasoning framework". It is unreasonable to conclude that the actual reasoning process is not robust based solely on errors induced during forced post-hoc justification. The framework measures how well a model can reverse-engineer an explanation for a pre-committed answer, which seems a different cognitive task than reasoning toward a conclusion.

2.The example provided in figure 1 actually hints at model resilience rather than complete fragility. The perturbed CoT output shows the model experiencing an "aha moment" where it explicitly catches its own inconsistency, for example "Wait, hold on" and "Let me check the question again". This indicates that, given a sufficiently long context window, the model may utilizes reflection to catch and potentially correct its post-hoc rationalizations. This may challenge the effectiveness of the proposed method.

**Requested Changes:**

Critical:

1.Provide a "CoT-then-Answer" baseline: The current approach forces an "answer-first, then reasoning" generation order. Could the authors either run baseline experiments using the standard 'CoT-then-Answer' format to prove this fragility exists in standard settings, or heavily revise their claims throughout the paper to clarify that they are strictly evaluating the robustness of post-hoc explanations?

2.Evaluate self-correction/reflection: Could the authors extend the maximum generation length to investigate whether the models can eventually self-correct and recover from these perturbations? Additionally, it would be helpful to include a discussion or justification on when these intermediate reflective corrections work or not.

---

> ### Author Response · Authors · 2026-03-25
> **Response**
>
> We thank the reviewer for their constructive feedback and provide detailed point-by-point responses below. Also, we incorporated them in the updated version of the manuscript.
>
> **1. Provide a "CoT-then-Answer" baseline**:
>
> We thank the reviewer for this important suggestion. We agree that evaluating standard CoT (CoT-then-Answer) prompting is necessary to determine whether the observed fragility is specific to the answer-conditioned setup. To address this, we implemented a CoT-then-Answer baseline with the same embedding-level perturbations. In this setting, it is impossible to explicitly fix or control the position of the generated final answer due to the autoregressive nature of LLMs. To approximate answer consistency, we identify the answer from the final portion of the generated sequence (e.g., the last quarter of tokens) and measure answer similarity based on this segment. Despite this constraint, we observe that reasoning instability persists in the standard CoT setting. Below are representative examples where small perturbations lead to clearly degraded or incoherent reasoning, even though the model attempts to follow a step-by-step format:
>
> - LLama3-8B:
>
> *Q: Carl has a cane that is half as long as he is tall.  Carl is one foot taller than his brother, Ned.  And Ned is two feet shorter than his cousin, Isabel.  If Isabel is 7 feet tall, how long is Carl's cane, in feet?*
>
> *A’: Let's think step by step. Isabel is://://.swing Isabel is the the.. the.  Carl is the taller than Ned by 7 feet.  Ned is 1 foot taller than Carl..  So Carl is 6 feet tall..  Carl's cane is half as long as he is tall..  So it is 3 feet long. The answer is 3.*
>
>
> - Zephyr-7B-beta:
>
> *Q: The vending machines sell chips for 40 cents and candy bars for 75 cents. George spent $5 and got 3 bags of chips and had 1% of his money left. How many candy bars did he buy?*
>
> *A’: Let's think step by step. George spent 5 dollars. 3 bags of chips cost 3 x 40 = 120 cents. So he spent 120 cents on chips. That leaves **5 - 120 = 380 cents** for candy bars and the remaining money. 1% of 5 dollars is 5 cents. So George has 380 - 5 = 375 cents left for candy bars. 375 cents is 5 candy bars at 75 cents each. So George bought 5 candy bars. The answer is 5.*
>
>
> Importantly, due to the autoregressive nature of LLMs, reasoning and answer generation are tightly coupled in the CoT-then-Answer setting. As a result, when the reasoning chain diverges, the final answer often becomes incorrect simultaneously. This makes it difficult to isolate reasoning-specific failures using answer correctness alone. To account for this, we adapt our attack success criterion in the CoT-then-Answer setting to focus on reasoning quality, evaluated using the same GPT-based judge. While this provides a practical way to assess reasoning instability under standard CoT prompting, we acknowledge that developing more precise methods to disentangle reasoning and answer generation in this setting remains an important direction for future work. Under this criterion, we observe non-trivial attack success rates across all models on GSM8K, as shown below:
>
>
> |        Model      |   ACC (C→A)  |   ACC (A→C)  |   UR (C→A)  |   UR (A→C)  |   SR (C→A)  |   SR (A→C)  |
> |:-----------------:|:------------:|:------------:|:-----------:|:-----------:|:-----------:|:-----------:|
> |   Llama-3-8B      | **75.6**  |   16.1       | **55.3** |   41.8      | **17.5** |   10.8      |
> |   Mistral-7B      | **45.4**  |   11.4       | **63.9** |   63.6      | **17.2** |   18.5      |
> |   Zephyr-7B-beta  | **43.8**  |   9.9        |   42.9      | **54.6** | **24.7** |   13.1      |
> |   Qwen2.5-7B      | **60.4**  |   24.8       | **60.3** |   46.8      | **6.3**  |   1.8       |
> |   DeepSeek-R1-7B  | **85.2**  |   22.5       | **80.8** |   59.3      | **7.7**  |   6.4       |

---

> > ### Author Response · Authors · 2026-03-25
> > **Response (2)**
> >
> > **2. Evaluate self-correction/reflection:**
> >
> > We observe that the DeepSeek-R1-7B model occasionally exhibits self-correction or reflection behaviors (e.g., revising intermediate reasoning steps). To evaluate whether such behaviors can mitigate reasoning fragility, we vary the generation length under embedding-level perturbations (256, 512, 1024 tokens). As shown below, increasing generation length leads to only modest and inconsistent changes in robustness. While the attack success rate (SR) slightly decreases in some cases (e.g., SingleEq), the overall effect is limited, and non-trivial attack success rates persist across all datasets. This indicates that longer reasoning and potential reflection do not fundamentally resolve the underlying instability.
> >
> >
> > | Dataset    | UR (256) | UR (512) | UR (1024) | SR (256) | SR (512) | SR (1024) |
> > |------------|----------|----------|-----------|----------|----------|-----------|
> > | SingleEq   | 20.6     | 16.9     | 12.6      | 6.1      | 4.0      | 3.5       |
> > | GSM8K      | 59.3     | 60.4     | 61.1      | 6.4      | 5.5      | 6.5       |
> > | StrategyQA | 10.0     | 18.5     | 23.1      | 33.1     | 27.8     | 26.0      |
> >
> > **3. Broader Impact Concerns**: Thanks for pointing out the potential concerns. We have added a broader impact section in the updated paper.

---

> > > ### Comment · Reviewer_sXF2 · 2026-04-02
> > >
> > > Thank you for your detailed response and for conducting these additional experiments. The inclusion of the CoT-first baseline and the analysis of reflection behaviors strengthen the paper and address my main concerns.

---

> > > > ### Author Response · Authors · 2026-04-02
> > > >
> > > > Thank you for recognizing our revisions. We are glad to hear that the additional experiments, including the CoT-first baseline and reflection analysis, address your main concerns. Please let us know if you have any additional questions.

---

### Author Response · Authors · 2026-03-25
**General Response**

We thank the reviewers for their constructive feedback. We have carefully revised the manuscript in response to their comments and would like to highlight the following key points.

1. Added a standard CoT (CoT-then-answer) baseline (all Reviewers).

2. Evaluated on a better judge GPT-4o and a larger model (Deepseek-14B) (Reviewers E7T9 and Tmzw).

3. Added a broader-impact section (Reviewers sXF2 and E7T9).

4. Evaluated self-correction/reflection (Reviewer sXF2).

5. Discussed selection bias (Reviewer Tmzw).

6. Added semantic analysis of token-level perturbation (Reviewer E7T9).

7. Tested on another benchmark (MATH) (Reviewer E7T9).

8. Reframed around robustness, not faithfulness (Reviewer Tmzw).

---

### Decision · Action_Editor_W6S7 · 2026-04-14

**Recommendation:** Accept with minor revision

**Additional Comments:**

1.Scope of claims: As the authors mentioned, reasoning and answer generation are coupled in the CoT-then-Answer setting due to self-regression nature. The paper should more explicitly and consistently qualify all claims about "decoupling" to be specific to the answer-conditioned setting throughout the manuscript.

2.Positioning relative to robustness literature: Given the reframing toward robustness, the related work section should more directly situate MATCHA against existing adversarial robustness work on NLP/reasoning tasks, and discuss what is genuinely new beyond showing that gradient-based attacks can degrade text generation quality.

3.Scaling discussion: The reduced attack success rates at 14B (near 1-2% on some datasets) deserve more explicit discussion.

**Audience:**

Yes

**Audience Explanation:**

The question of whether LLM reasoning chains reflect genuine computation or are loosely coupled rationalizations is directly relevant to the interpretability, robustness, and alignment communities. The MATCHA framework provides a practical diagnostic tool, and the empirical findings, including transferability to closed-source models and persistence of fragility under scaling, are informative.

**Claims And Evidence:**

Yes

**Claims Explanation:**

The revised submission provides sufficient evidence to support its core claims, appropriately reframed around CoT robustness rather than causal faithfulness. The authors introduce MATCHA, a framework that demonstrates a meaningful gap between answer robustness and rationale robustness under small adversarial perturbations, validated across multiple models, datasets, and evaluation settings. In response to reviewer concerns, the authors added: (i) a CoT-then-Answer baseline confirming that reasoning instability persists beyond the answer-first setup, (ii) stronger judge validation and human annotation showing consistent trends, (iii) rule-based arithmetic verification as an independent signal, (iv) additional benchmark coverage, and (v) a scaling comparison (7B vs. 14B). The finding that rationales are substantially more fragile than final answers under targeted perturbation is consistently supported across these evaluations.

However, the standard CoT experiments show that perturbations degrade both reasoning and the final answer simultaneously, which complicates the decoupling interpretation. The experimental design — conditioning reasoning generation on a pre-committed answer — is by construction a post-hoc rationalization task. The authors' reframing toward robustness is appreciated, but does not fully resolve the concern that the observed fragility may be a property of forced rationalization rather than of CoT reasoning more broadly. The authors should explicitly clarify in the revision that the Decoupling Hypothesis, as supported by the current evidence, applies specifically to post-hoc rationalization settings.

---

> ### Author Response · Authors · 2026-05-14
> **Thank you and uploaded the camera ready version**
>
> Thank you for the thoughtful and constructive feedback. We have carefully addressed all three points in the final version of the paper. Specifically, we clarified the scope of our claims regarding “decoupling” to emphasize the answer-conditioned setting consistently, expanded the related work discussion to better position MATCHA within the broader adversarial robustness literature on NLP and reasoning tasks, and added a more explicit discussion on scaling behavior, particularly the reduced attack success rates observed for 14B models. We sincerely appreciate these suggestions, which helped improve both the clarity and positioning of the work. Please feel free to let us know if there are any other issues we need to address further.